# Gradient Rewiring for Editable Graph Neural Network Training

**Zhimeng Jiang**[1], **Zirui Liu**[2], **Xiaotian Han**[3], **Qizhang Feng**[1], **Hongye Jin**[1],
**Qiaoyu Tan**[4], **Kaixiong Zhou**[5], **Na Zou**[6], **Xia Hu**[7]
[1]Texas A&M University, [2]University of Minnesota, [3]Case Western Reserve University,
[4]NYU Shanghai, [5]North Carolina State University, [6]University of Houston, [7]Rice University

## Abstract

Deep neural networks are ubiquitously adopted in many applications, such as computer vision, natural language processing, and graph analytics. However, well-trained neural networks can make prediction errors after deployment as the world changes. *Model editing* involves updating the base model to correct prediction errors with less accessible training data and computational resources. Despite recent advances in model editors in computer vision and natural language processing, editable training in graph neural networks (GNNs) is rarely explored. The challenge with editable GNN training lies in the inherent information aggregation across neighbors, which can lead model editors to affect the predictions of other nodes unintentionally. In this paper, we first observe the gradient of cross-entropy loss for the target node and training nodes with significant inconsistency, which indicates that directly fine-tuning the base model using the loss on the target node deteriorates the performance on training nodes. Motivated by the gradient inconsistency observation, we propose a simple yet effective Gradient Rewiring method for Editable graph neural network training, named **GRE**. Specifically, we first store the anchor gradient of the loss on training nodes to preserve the locality. Subsequently, we rewire the gradient of the loss on the target node to preserve performance on the training node using anchor gradient. Experiments demonstrate the effectiveness of GRE on various model architectures and graph datasets in terms of multiple editing situations. The source code is available at https://github.com/zhimengj0326/Gradient_rewiring_editing.

## 1 Introduction

Graph Neural Networks (GNNs) have demonstrated exemplary performance for graph learning tasks, such as recommendation, link prediction, molecule property analysis [1, 2, 3, 4, 5, 6, 7]. With message passing, GNNs learn node representations by recursively aggregating the neighboring nodes' representations. Once trained, GNN models are deployed to handle various high-stake tasks, such as credit risk assessment in financial networks [8] and fake news detection in social networks [9]. However, the impact of erroneous decisions in such influential applications can be substantial. For instance, misplaced credit trust in undetected fake news can lead to severe financial loss.

An ideal approach to tackle such errors should possess the following properties: 1) the ability to *rectify* severe errors in the model's predictions, 2) the capacity to *generalize* these corrections to other similar instances of misclassified samples, and 3) the ability to *preserve* the model's prediction accuracy for all other unrelated inputs. To achieve these goals, various model editing frameworks have been developed to rectify errors by dynamically adjusting the model's behavior when errors are detected [10, 11]. The core principle is to implement minimal changes to the model to correct the error while keeping the rest of the model's behavior intact. However, model editing is not a simple

38th Conference on Neural Information Processing Systems (NeurIPS 2024).

plug-and-play solution. These frameworks often require an additional training phase to prepare for editing before they can be used effectively for editing [10, 11, 12, 13]. Although model editing techniques have shown significant utility in computer vision and language models, there is rare work focused on rectifying critical errors in graph data. The unique challenge arises from the inherent message-passing mechanism in GNNs when edits involve densely interconnected nodes [14, 15]. Specifically, editing the behavior of a single node can unintentionally induce a ripple effect, causing changes that propagate throughout the entire graph. [14] theoretically and empirically demonstrate the complexity of editing GNNs through the lens of the loss landscape of the Kullback-Lieber divergence between the pre-trained node features and edited final node embeddings. Moreover, a simple yet effective model structure, named EGNN, is proposed with stitched peer multi-layer perception (MLP), where only the stitched MLP is trained during model editing.

In this work, we investigate the model editing problem for GNNs from a *brand-new gradient perspective*, which is compatible with existing work [14]. Specifically, we first found a considerable inconsistency between the gradients of the cross-entropy loss for the target node and the training nodes for GNNs. Such inconsistency implies that direct fine-tuning of the base model using the loss of the target node can lead to a deterioration in the performance on the training nodes. Motivated by the above observation, we propose a simple yet effective Gradient Rewiring method for Editable graph neural network training, named **GRE**. Specifically, we first calculate and store the anchor gradient of the loss on the training nodes. This anchor gradient represents the original learning direction that we wish to preserve. Then, during the editing process, we adjust the gradient of the loss on the target node based on the stored anchor gradient. This adjustment, or "rewiring", ensures that the changes made to the target node do not adversely affect the performance on the training nodes. Experiments demonstrate the effectiveness of our proposed method for various model structures and graph datasets. Moreover, the proposed method is compatible with the existing EGNN baseline and further improves the performance.

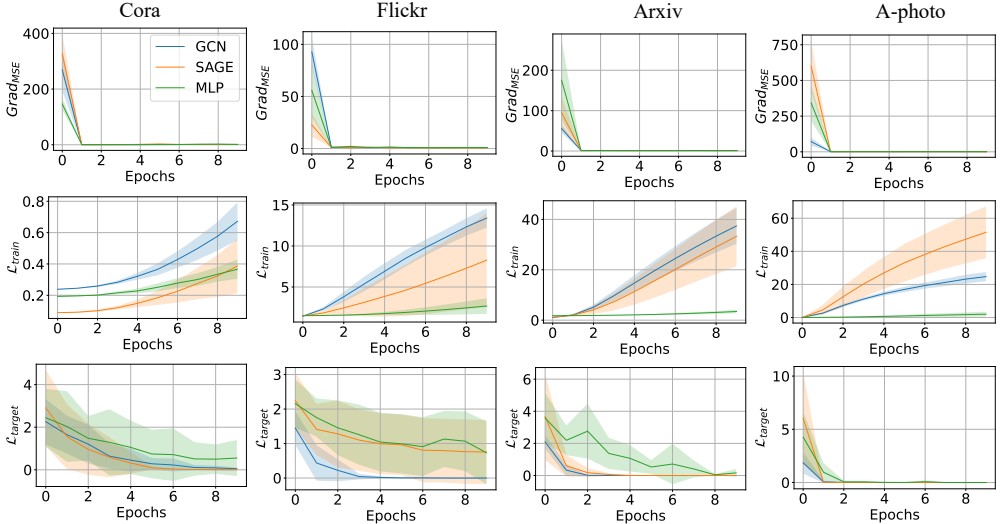

Figure 1: (a) Top: RMSE distance between the gradients of cross-entropy loss over training datasets and over the targeted sample for different architectures. (b) Middle: Cross-entropy loss over training datasets when the model is updated using target loss. (c) Bottom: Cross-entropy loss over the targeted sample when the model is updated using target loss.

## 2 Preliminary and Related Work

We first introduce the notations used throughout this paper. A graph is given by $\mathcal{G} = (\mathcal{V}, \mathcal{E})$, where $\mathcal{V} = (v_1, \cdots, v_N)$ is the set of nodes indexed from 1 to $n$, and $\mathcal{E} = (e_1, \cdots, e_m) \subseteq \mathcal{V} \times \mathcal{V}$ is the set of edges. $n = |\mathcal{V}|$ and $m = |\mathcal{E}|$ are the numbers of nodes and edges, respectively. Let $\boldsymbol{X} \in \mathbb{R}^{n \times d}$ be the node feature matrix, where $d$ is the dimension of node features. $\boldsymbol{A} \in \mathbb{R}^{n \times n}$ is the graph adjacency matrix, where $\boldsymbol{A}_{i,j} = 1$ if $(v_i, v_j) \in \mathcal{E}$ else $\boldsymbol{A}_{i,j} = 0$. $\tilde{\boldsymbol{A}} = \tilde{\boldsymbol{D}}^{-\frac{1}{2}}(\boldsymbol{A} + \boldsymbol{I})\tilde{\boldsymbol{D}}^{-\frac{1}{2}}$ is the normalized

adjacency matrix, where $\tilde{\boldsymbol{D}}$ is the degree matrix of $\boldsymbol{A} + \boldsymbol{I}$. The node label is defined as $y_i$ for node $v_i$. We consider node classification tasks with $C$ classes in this paper.

## 2.1 Graph Neural Networks

Graph Neural Networks have been successfully applied across various domains/tasks, including knowledge graphs [16, 17], graph condensation [18, 19, 20], event extraction [21], and entity relation tasks [22]. Most graph neural networks follow a neighborhood aggregation procedure to learn node representation via propagating representations of neighbors and then follow up with feature transformation [23]. The $l$-th layer of graph neural networks is given by:

$$\mathbf{a}_i^{(l)} = \text{PROPAGATION}^{(l)}\Big(\big\{\mathbf{x}_i^{(l-1)}, \mathbf{x}_j^{(l-1)} | j \in \mathcal{N}_i\big\}\Big),$$

$$\mathbf{x}_i^{(l)} = \text{TRANSFORMATION}^{(l)}\Big(\mathbf{a}_i^{(l)}\Big),$$

where $\mathbf{x}_i^{(l)}$ is the representation of node $v_i$ at $l$-th layer and $x_i^{(0)}$ is initialized as node feature $\mathbf{x}_i$, i..e, the $i$-th row at node feature matrix $\boldsymbol{X}$. Many GNNs, such as GCN [24], GraphSAGE [25], and GAT [26], can be defined under this computation paradigm via adopting the different propagation and transformation operations. For example, the $l$-th layer in GCN can be defined as:

$$\boldsymbol{X}^{(l)} = \sigma(\tilde{\boldsymbol{A}}\boldsymbol{X}^{(l-1)}\boldsymbol{W}^{(l)}), \tag{1}$$

where $\boldsymbol{X}^{(l)} \in \mathbb{R}^{n \times d}$ and $\boldsymbol{X}^{(l-1)} \in \mathbb{R}^{n \times d}$ are the node representation matrix containing the $\boldsymbol{h}_v$ for each node $v$ at the layer $l$ and layer $l-1$, respectively. $\boldsymbol{W}^{(l)} \in \mathbb{R}^{d \times d}$ is a layer-specific trainable weight matrix, and $\sigma(\cdot)$ is a non-linear activation function (e.g., ReLU).

## 2.2 Model Editing

Model editing aims to modify a base model's responses for a misclassified sample $x_{tg}$ and its analogs. This is typically achieved by fine-tuning the model using only a single pair of input $x_{tg}$ and the desired output $y_{tg}$, while preserving the model's responses to unrelated inputs [10, 11, 12, 14]. Our contribution lies in the novel application of model editing to graph data, a domain where misclassifications on a few pivotal nodes can trigger substantial financial losses, fairness issues, or even the propagation of adversarial attacks. Consider the scenario of node classification where a well-trained GNN incorrectly predicts a particular node. Model editing can be employed to rectify this erroneous prediction. By leveraging the node's characteristics and the desired label, the model can be updated to correct such behavior. The ideal outcome of model editing is twofold: first, the updated model should correctly predict the specific node and its similar instances; second, the model should maintain its original behavior on unrelated inputs. It is important to note that some model editors require a preparatory training phase before they can be applied effectively [10, 13, 12, 11]. This crucial step ensures that the model editing process is both precise and effective in its application.

# 3 Methodology

In this section, we first provide the preliminary experimental results as the motivation to rewire gradients for model editing. Subsequently, we propose our gradient rewiring method for editable graph neural networks training (GRE) and an advanced version (GRE+) to improve the effectiveness of model editing, respectively.

## 3.1 Motivation

In the preliminary experiments, we first pre-train GCN, GraphSAGE, and MLP on the training dataset $\mathcal{V}_{train}$ (e.g., Cora, Flickr, ogbn-arxiv, and Amazon Photo datasets) using cross-entropy loss. Subsequently, we find the misclassified samples in the validation dataset and randomly select one sample as the target sample $(\mathbf{x}_{tg}, y_{tg})$. During the model editing, we update the pre-trained model using cross-entropy loss over the target sample using gradient descent, i.e., *the models are trained inductively*. Following previous work [10, 12, 14], we perform 50 independent edits and report the averaged metrics.

It is well-known that model editing incurs training performance degradation [11, 10, 12] [1] for many model architectures. To deeply delve into the underlying reason, we investigate performance degradation from a model gradient perspective. We further define the training loss as $\mathcal{L}_{train} = \frac{1}{|\mathcal{V}_{train}|} \sum_{i \in \mathcal{V}_{train}} CE(f_\theta(\mathbf{x}_i), y_i)$, where $f_\theta(\cdot) \in \mathbb{R}^C$ is a prediction model parameterized with $\theta \in \mathbb{R}^L$, $C$ and $L$ are the number of classes and model parameters, $CE(\cdot, \cdot)$ is the cross-entropy loss, the target loss is given by $\mathcal{L}_{tg} = CE(f_\theta(\mathbf{x}_{tg}), y_{tg})$. For example, model $f_\theta(\cdot)$ can be instantiated by GNNs with the number of layers defined in Eq. (1) or a simple MLP. For model editing, the gradient for training and target loss is given by $g_{train} = \frac{\partial \mathcal{L}_{train}}{\partial \theta} \in \mathbb{R}^L$ and $g_{tg} = \frac{\partial \mathcal{L}_{tg}}{\partial \theta} \in \mathbb{R}^L$, respectively. To investigate why the model editing leads to training performance degradation, we use gradient RMSE (Root-Mean-Squared-Error), i.e., $\text{Grad}_{RMSE} = \sqrt{\|g_{train} - g_{tg}\|_2^2}$, to measure the model editing discrepancy for training datasets and target sample.

The model editing curves for gradient RMSE [2], training loss, and target loss across various model architectures (GCN, GraphSAGE, and MLP) are shown in Figure 1. Although the gradient RMSE for training datasets and target sample is close to 0, the model parameters demonstrate significant inconsistent behavior in terms of training loss due to large gradient discrepancy in the initial editing stage. We observe that: 1) Even though the target loss decreases during model editing, the training loss increases significantly. 2) The increasing rates of training loss for GCN and GraphSAGE are significantly higher than that of MLP. The above observations imply that editing training for graph neural networks is more challenging due to higher gradient discrepancy between the training dataset and the target sample.

### 3.2 Gradient Rewiring Approach

Preliminary results show a high discrepancy in training loss and target loss for GNNs, which implies that the vanilla model editing hampers the performance on the overall training dataset and thus results in a high accuracy drop for node classification tasks. Therefore, we aim to tackle the training dataset performance degradation from the gradient rewiring approach.

**GRE** We propose a simple yet effective gradient rewiring approach for editable graph neural network training, named GRE. We first formulate a constrained optimization problem to *regulate* model editing and then solve the constrained optimization problem via gradient rewiring.

Model editing aims to correct the prediction for the target sample while maintaining the prediction accuracy on the training nodes. The objective function focuses on minimizing the loss at the target node. To preserve the predictions on the training nodes, we introduce two constraints: (1) the training loss should not exceed its value prior to model editing (see Eq. (3)); and (2) the differences in model predictions after editing should remain within a predefined range (see Eq. (4)). Define $\theta_0$ and $\theta'$ as the model parameters before and after model editing. Then we have the following constrained optimization problem:

$$\min_{\theta} \quad \mathcal{L}_{tg}\big(f_\theta(\mathbf{x}_{tg}), y_{tg}\big) \tag{2}$$

$$\text{s.t.} \quad \mathcal{L}_{train}\big(f_{\theta'}, \mathcal{V}_{train}\big) \leq \mathcal{L}_{train}\big(f_{\theta_0}, \mathcal{V}_{train}\big) \tag{3}$$

$$\|\frac{1}{|\mathcal{V}_{train}|} \sum_{i \in \mathcal{V}_{train}} f_{\theta'}(\mathbf{x}_i) - f_{\theta_0}(\mathbf{x}_i)\|^2 \leq \delta', \tag{4}$$

where $\theta$ and $\theta'$ represent the model parameters before and after model editing, respectively, the hyperparameter $\delta'$ represents the maximum average prediction difference on training nodes. Notice that the model parameters update adopts gradient descent using target loss without any constraints, i.e., $\theta' = \theta_0 - \alpha g_{tg}$, where $\alpha$ is step size in model editing. The key idea of our proposed solution is to rewire gradient $g_{tg}$ as $g$, which is obtained by satisfying the involved constraints. Note that the model editing usually corrects the model prediction on the target sample within a few steps, i.e. there are no significant model parameter differences, thus we adopt Taylor expansion to tackle such constrained

---

[1]The reason for focusing on the training set is that during model editing, we can only use the training set and not the test set.

[2]There is no variance for gradient estimation since gradient calculation is based on backpropagation. The large variance in model performance and gradient discrepancy derives from the randomly selected target node.

optimization problem. For target loss $\mathcal{L}_{tg}$, we can approximate it as:

$$
\begin{aligned}
\mathcal{L}_{tg}\big(f_{\theta'}(\mathbf{x}_{tg}), y_{tg}\big) &\approx \mathcal{L}_{tg}\big(f_{\theta_0}(\mathbf{x}_{tg}), y_{tg}\big) + g_{tg}^\top(\theta' - \theta_0) \\
&= \mathcal{L}_{tg}\big(f_{\theta_0}(\mathbf{x}_{tg}), y_{tg}\big) - \alpha g_{tg}^\top g.
\end{aligned}
\tag{5}
$$

To optimize the objective function Eq. (2), it is easy to conclude that the gradient cosine similarity $g_{tg}^\top g$ should be maximized. Given the gradient before/after model editing is fixed, the maximization of gradient cosine similarity $g_{tg}^\top g$ is equivalent to the minimization of $\|g_{tg} - g\|^2$. To satisfy Eq. (3), we also adopt Taylor expansion on $\mathcal{L}_{train}$ and it is easy to obtain that the gradient cosine similarity should be positive, i.e., $g_{tg}^\top g \geq 0$. As for the constraint in Eq. (4), similarly, a Taylor expansion is used to express the relationship between the model predictions before and after the model editing, as follows:

$$
f_{\theta'}(\mathbf{x}_i) \approx f_{\theta_0}(\mathbf{x}_i) + \frac{\partial f_{\theta_0}(\mathbf{x}_i)}{\partial \theta}^\top (\theta' - \theta) = f_{\theta_0}(\mathbf{x}_i) - \frac{\partial f_{\theta_0}(\mathbf{x}_i)}{\partial \theta}^\top \cdot \alpha g,
\tag{6}
$$

Therefore, we can obtain the following approximation on Eq. (4):

$$
\|\frac{1}{|\mathcal{V}_{train}|} \sum_{i \in \mathcal{V}_{train}} f_{\theta'}(\mathbf{x}_i) - f_{\theta_0}(\mathbf{x}_i)\|^2 \approx \|\hat{g}_{train}^\top(-\alpha g)\|^2 \leq \delta',
\tag{7}
$$

where gradient for a model prediction is defined as $\hat{g}_{train} = \frac{\partial \frac{1}{|\mathcal{V}_{train}|} \sum_{i \in \mathcal{V}_{train}} f_{\theta_0}(\mathbf{x}_i)}{\partial \theta}\big|_{\theta = \theta_0} \in \mathbb{R}^{L \times C}$. Therefore, the model prediction difference constraint can be transformed into $\|\hat{g}_{train}^\top g\|^2 \leq \|\hat{g}_{train}\|_{spect}^2 \|g\|^2 \leq \delta$, where $\|\cdot\|_{spect}$ represents matrix spectrum norm and $\|\hat{g}_{train}\|_{spect}$ is fixed in model editing, and $\delta = \frac{\delta'}{\alpha^2}$. In a nutshell, our goal is to correct the target sample (i.e., minimize $\|g_{tg} - g\|^2$) and minimize gradient discrepancy for model prediction among training dataset and target sample (i.e., $\|g\|^2$), while guaranteeing non-increased training loss (i.e., $g_{train}^\top g \geq 0$). The original constraint optimization problem is simplified as gradient rewiring, i.e.,

$$
\min_g \frac{1}{2}\|g - g_{tg}\|^2 + \frac{\lambda}{2}\|g\|^2 = \min_g \frac{1+\lambda}{2} g^\top g - g_{tg}^\top g + \frac{1}{2} g_{tg}^\top g_{tg} \quad \text{s.t.} \quad g_{train}^\top g \geq 0, \tag{8}
$$

where $\lambda \geq 0$ is the hyperparameter to control the balance between target sample correction and gradient discrepancy for model prediction. It is easy to obtain that Eq.(8) is a quadratic program (QP) in $L$-variables (the number of model parameters is usually high in neural networks). Fortunately, we can effectively solve this problem in the dual space via transforming as a smaller QP problem with only one variable $v$ [27], where the relation between primal and dual variable is $g_{train}v - (1+\lambda)g = -g_{tg}$. Then we have the following problem:

$$
\min_v \frac{(1+\lambda)^{-1}}{2}(g_{train}v + g_{tg})^\top (g_{train}v + g_{tg}) \quad \text{s.t. } v \geq 0. \tag{9}
$$

It is easy to obtain the optimal dual variable $v^* = -\min\{\frac{g_{train}^\top g_{tg}}{g_{train}^\top g_{train}}, 0\}$ and the optimal rewired gradient $g^* = (1+\lambda)^{-1}(g_{tg} - v^* g_{train})$. In other words, the gradient rewiring procedure is quite simple: for the gradient of the target loss $g_{tg}$, reduce its projection component on $g_{train}$ and then scale it by $(1+\lambda)^{-1}$.

Additionally, we highlight that the gradient for training loss $g_{train}$ must be stored before model editing. In this way, gradient rewiring can be conducted to remove the harmful gradient component on target loss that increases training loss. Since shallow GNNs model performs well in practice [28], the model size of GNNs is small and the memory cost $O(L)$ for storing anchor gradient is negligible.

**GRE+**  In GRE, the training loss after model editing is required not to be larger than that before model editing. However, it is still possible that the training loss on specific sub-training sets performs worse after model editing. At the same time, the training loss for the whole training dataset, after model editing, is on par with or even lower than that of before editing. To tackle this issue, we proposed an advanced gradient rewiring approach, named GRE+, via applying loss constraint on multiple disjoint sub-training sets. Specifically, we split training dataset $\mathcal{V}_{train}$ into $K$ sub-training sets $\{\mathcal{V}_{train}^1, \mathcal{V}_{train}^2, \cdots, \mathcal{V}_{train}^K\}$. Similarly, we define $g_{train}^k = \frac{\partial \mathcal{L}_{train}^k}{\partial \theta} \in \mathbb{R}^L$, where $\mathcal{L}_{train}^k = \frac{1}{|\mathcal{V}_{train}^k|} \sum_{i \in \mathcal{V}_{train}^k} CE(f_\theta(\mathbf{x}_i), y_i)$.

Following the derivative clue in GRE, we can replace the training loss constraint on the whole training dataset with multiple training loss constraints on training subsets, and obtain the advanced gradient rewiring approach as follows:

$$\min_g \frac{1}{2}\|g - g_{tg}\|^2 + \frac{\lambda}{2}\|g\|^2 = \min_g \frac{1+\lambda}{2}g^\top g - g_{tg}^\top g + \frac{1}{2}g_{tg}^\top g_{tg}$$
$$\text{s.t. } (g_{train}^k)^\top g \geq 0, \text{ for any } 1 \leq k \leq K. \tag{10}$$

Notice that Eq.(10) is a quadratic program (QP) in $L$-variables (the number of model parameters are usually high in neural networks), and we can effectively solve this problem in the dual space via transforming as a smaller QP problem with only $K$ variables $\mathbf{v} \in \mathbb{R}^K$ [27]. Define gradient matrix as $\boldsymbol{G} = [g_{train}^1, g_{train}^2, \cdots, g_{train}^K]^\top \in \mathbb{R}^{K \times L}$, then the relation between primal and dual variable is given by $\boldsymbol{G}\mathbf{v} - (1+\lambda)g = -g_{tg}$. The original optimization problem can be transformed into the following dual problem:

$$\min_{\mathbf{v}} \quad \frac{(1+\lambda)^{-1}}{2}\mathbf{v}^\top \boldsymbol{G}\boldsymbol{G}^\top \mathbf{v} + (1+\lambda)^{-1}g_{tg}^\top \boldsymbol{G}^\top \mathbf{v} + (1+\lambda)^{-1}g_{tg}^\top g_{tg}$$
$$\text{s.t.} \quad \mathbf{v}_k \geq 0, \text{ for any } 1 \leq k \leq K. \tag{11}$$

The dual problem is a QP with $K \ll L$ variables, and we usually consider the value of $K$ to be smaller than 5 in practice. Once we tackle dual QP problem (11) for $\mathbf{v}^*$, we can recover the rewired gradient as $g = (1+\lambda)^{-1}(\boldsymbol{G}\mathbf{v} + g_{tg})$. Similarly, the gradient for training loss $g_{train}^k$, where $1 \leq k \leq K$, is required to be stored before model editing. The corresponding memory cost is given by $O(KL)$.

Table 1: The results on four small-scale datasets after applying one single edit. The reported number is averaged over 50 independent edits. **SR** is the edit success rate, **Acc** is the test accuracy after editing, and **DD** are the test drawdown, respectively. "OOM" is the out-of-memory error. The best/second-best results are highlighted in **boldface**/underlined, respectively.

| | Editor | Cora | | | A-computers | | | A-photo | | | Coauthor-CS | | |
|---|---|---|---|---|---|---|---|---|---|---|---|---|---|
| | | Acc↑ | DD↓ | SR↑ | Acc↑ | DD↓ | SR↑ | Acc↑ | DD↓ | SR↑ | Acc↑ | DD↓ | SR↑ |
| MLP | GD | 68.15±0.33 | 3.85±0.33 | 0.98 | **73.22±0.48** | **6.78±0.48** | 1.00 | **83.19±0.91** | **6.81±0.91** | 1.00 | 93.59±0.05 | 0.41±0.05 | 1.00 |
| | ENN | 37.16±3.80 | 52.24±4.76 | 1.00 | 15.51±10.99 | 72.36±10.87 | 1.00 | 16.71±14.81 | 77.07±15.20 | 1.00 | 4.94±3.78 | 89.43±3.34 | 1.00 |
| | GRE | 69.41±0.44 | 2.59±0.44 | 0.96 | 61.21±1.26 | 18.79±1.26 | 1.00 | 73.56±1.41 | 16.44±1.41 | 1.00 | 93.27±0.09 | 0.73±0.09 | 1.00 |
| | GRE+ | **71.19±0.28** | **0.61±0.28** | 0.96 | 61.27±1.15 | 18.73±1.15 | 1.00 | 78.26±1.15 | 11.74±1.15 | 1.00 | **93.73±0.07** | **0.27±0.07** | 1.00 |
| GCN | GD | 84.37±5.84 | 5.03±6.40 | 1.00 | 44.78±22.41 | 43.09±22.32 | 1.00 | 28.70±21.26 | 65.08±20.13 | 1.00 | 91.07±3.23 | 3.30±2.22 | 1.00 |
| | ENN | 37.16±3.80 | 52.24±4.76 | 1.00 | 15.51±10.99 | 72.36±10.87 | 1.00 | 16.71±14.81 | 77.07±15.20 | 1.00 | 4.94±3.78 | 89.43±3.34 | 1.00 |
| | GRE | 84.98±0.47 | 4.02±0.47 | 0.96 | 46.28±3.47 | 51.72±3.47 | 0.98 | 35.88±2.26 | 58.12±2.26 | 0.99 | 89.46±0.29 | 4.54±0.29 | 1.00 |
| | GRE+ | **88.84±0.35** | **0.56±0.35** | 0.98 | **47.75±0.45** | **40.25±0.45** | 1.00 | **50.13±1.36** | **43.87±1.36** | 1.00 | **91.99±0.30** | **2.01±0.30** | 1.00 |
| Graph-SAGE | GD | 82.06±4.33 | 4.54±5.32 | 1.00 | 21.68±20.98 | 61.15±20.33 | 1.00 | 38.98±30.24 | 55.32±29.35 | 1.00 | 90.15±5.58 | 5.01±5.32 | 1.00 |
| | ENN | 33.16±1.45 | 53.44±2.23 | 1.00 | 16.89±16.98 | 65.94±16.75 | 1.00 | 15.06±11.92 | 79.24±11.25 | 1.00 | 13.71±2.73 | 81.45±2.11 | 1.00 |
| | GRE | 83.64±0.20 | 3.36±0.20 | 1.00 | 20.11±2.30 | 62.89±2.30 | 0.96 | 41.96±1.57 | 52.04±1.57 | 0.98 | 91.07±0.44 | 3.93±0.44 | 1.00 |
| | GRE+ | **86.59±0.07** | **0.41±0.07** | 1.00 | **22.23±1.60** | **60.77±1.60** | 0.97 | **44.05±0.83** | **50.32±0.83** | 1.00 | **91.75±0.43** | **3.25±0.43** | 1.00 |
| EGNN-GCN | GD | 87.58±0.31 | 1.42±0.31 | 1.00 | 87.27±0.14 | 0.73±0.14 | 0.78 | 93.24±0.59 | 0.76±0.59 | 0.77 | 93.99±0.02 | 0.01±0.02 | 0.91 |
| | GRE | 87.47±0.41 | 1.53±0.41 | 1.00 | 83.38±1.20 | 4.62±1.20 | 0.87 | 88.01±1.20 | 5.99±1.20 | 0.86 | 93.92±0.07 | 0.08±0.07 | 0.94 |
| | GRE+ | **88.99±0.21** | **0.05±0.21** | 1.00 | **88.10±1.21** | **0.51±1.21** | 1.00 | **94.22±0.98** | **−0.21±0.98** | 1.00 | **94.32±0.06** | **−0.32±0.06** | 1.00 |
| EGNN-SAGE | GD | 85.05±0.11 | 0.95±0.11 | 1.00 | 85.93±0.08 | 0.07±0.08 | 0.90 | 93.87±0.20 | 0.13±0.20 | 0.81 | 95.0±0.01 | 0.00±0.01 | 0.99 |
| | GRE | 84.79±0.19 | 1.21±0.19 | 1.00 | 81.94±1.71 | 4.06±1.71 | 0.96 | 88.55±1.19 | 5.45±1.19 | 0.95 | 94.85±0.05 | 0.15±0.05 | 1.00 |
| | GRE+ | **86.24±1.43** | **−0.24±1.43** | 1.00 | **85.97±0.83** | **−0.16±0.83** | 1.00 | **94.07±0.03** | **−0.07±0.03** | 0.98 | **95.07±0.03** | **−0.07±0.03** | 1.00 |

## 4 Experiments

In this section, we conduct experiments to evaluate the effectiveness of our proposed GRE and GRE+, with the goal of answering the following three research questions. **RQ1:** Can the proposed solution correct the wrong model prediction with a lower accuracy drop after model editing in the independent and sequential editing setting? **RQ2:** What's the tradeoff performance between accuracy drop and success rate in the independent editing setting? **RQ3:** How sensitive are the proposed GRE and GRE+ methods to the key hyperparameter $\lambda$?

### 4.1 Experimental Setting

We follow the standard experimental setting for GNNs [14]. Specifically, we first randomly split the train/validation/test dataset. Then, we ensure that each class has 20 samples in the training and 30 samples in the validation sets. The remaining samples are used for the test set. The target node is randomly selected 50 times from the validation set and the well-trained model makes the wrong prediction. The average model editing performance (e.g., success rate, drawdown) is reported for evaluation.

**Datasets and Models.**    In our experiments, we utilize a selection of eight graph datasets from diverse domains, split evenly between small-scale and large-scale datasets. The small-scale datasets include Cora, A-computers [29], A-photo [29], and Coauthor-CS [29]. On the other hand, the large-scale datasets encompass Reddit [25], Flickr [2], *ogbn-arxiv* [3], and *ogbn-products* [3]. Note that our approach is based on gradient rewiring, which is orthogonal to model architectures. We adopt two prevalent models GCN [24] and GraphSAGE [25], where both of them are trained with the entire graph at each step. We evaluate our method under the **inductive setting**, which means the model is trained on a subgraph containing only the training node, and evaluated on the whole graph.

**Baselines.**    Our methods are evaluated against three notable baselines: the traditional gradient descent editor (GD), the Editable Neural Network editor (ENN) [10], and editable training for GNNs[14]. [3]  The GD editor is a straightforward application using gradient descent on the target loss with respect to the GNNs model parameters until the desired prediction outcome is achieved. ENN adopts a different approach by initially training the GNN parameters for a few steps to prime the model for subsequent edits. After this preparatory phase, ENN, like GD, applies the gradient descent on the parameters of GNN until the correct prediction is attained. EGNN [14] stitches a peer MLP and only trains MLP during model editing. Note that our method is compatible with EGNN, and different GNN architectures integrated with EGNN (e.g., EGNN-GCN, EGNN-GraphSAGE) are treated as distinct architectures.

**Independent, sequential, and batch editing.**    All independent, sequential, and batch editing processes involve well-trained GNN models using training datasets, with target samples randomly selected multiple times from misclassified instances in the validation dataset. The key differences lie in the base model that needs to be edited. For independent editing, the same well-trained model using the training datasets is edited multiple times. In contrast, for sequential editing, the model is edited iteratively, with each editing step using the previously edited model from the last target sample, incorporating both the training datasets and partial samples from the validation dataset. For batch editing, all batched samples are edited simultaneously in one editing process. [4]

**Evaluation Metrics.**    Consistent with preceding studies [10, 12, 11], we assess the effectiveness of the various methods using two primary metrics: (1) **Accuracy (Acc)**: We use accuracy for the test dataset to evaluate the effectiveness after model editing. (2) **DrawDown (DD)**: This metric measures the mean absolute difference in test accuracy before and after model editing. A lower drawdown value signifies a superior editor locality. (3) **Success Rate (SR)**: This metric evaluates the proportion of edits in which the editor successfully amends the model's prediction. Both metrics offer a different perspective on the effectiveness of the editing process.

Table 2: The results on four large-scale datasets after applying one single edit. "OOM" is the out-of-memory error. The best/second-best results are highlighted in **boldface**/underlined, respectively. The results for more backbones (e.g., MLP, EGNN-GCN, EGNN-SAGE) are in Appendix D.1.

| | Editor | Flickr | | | Reddit | | | ogbn-arxiv | | | ogbn-products | | |
|---|---|---|---|---|---|---|---|---|---|---|---|---|---|
| | | Acc↑ | DD↓ | SR↑ | Acc↑ | DD↓ | SR↑ | Acc↑ | DD↓ | SR↑ | Acc↑ | DD↓ | SR↑ |
| GCN | GD | 13.95±11.00 | 37.25±10.20 | 1.00 | 75.20±12.30 | 20.32±11.30 | 1.00 | 23.71±16.90 | 46.50±14.90 | 1.00 | 53.29±0.94 | 20.71±0.94 | 1.00 |
| | ENN | **25.82**±14.90 | **25.38**±16.90 | 1.00 | 11.16±5.10 | 84.36±3.10 | 1.00 | 16.59±7.70 | 53.62±6.70 | 1.00 | OOM | OOM | OOM |
| | GRE | 17.36±1.50 | 33.64±1.50 | 0.98 | 24.74±1.92 | 45.26±1.92 | 1.00 | 77.84±1.16 | 18.16±1.16 | 1.00 | 53.99±0.60 | 20.01±0.60 | 1.00 |
| | GRE+ | 22.9±0.67 | 28.1±0.67 | 0.97 | 34.15±1.33 | 35.85±1.33 | 1.00 | **80.61**±1.10 | **15.39**±1.10 | 1.00 | **57.43**±1.30 | **16.89**±1.30 | 1.00 |
| Graph-SAGE | GD | 17.16±12.20 | 31.88±12.20 | 1.00 | **55.85**±22.50 | 40.71±20.30 | 1.00 | 19.07±14.10 | 36.68±10.10 | 1.00 | 62.16±2.10 | 4.38±2.10 | 1.00 |
| | ENN | 28.73±5.60 | 20.31±5.60 | 1.00 | 5.88±3.90 | 90.68±4.30 | 1.00 | 8.14±8.60 | 47.61±7.60 | 1.00 | OOM | OOM | OOM |
| | GRE | 20.69±1.62 | 28.31±1.62 | 0.99 | 21.93±0.94 | 47.07±0.94 | 1.00 | 47.16±1.22 | 48.84±1.22 | 1.00 | 61.96±1.02 | 4.58±1.02 | 1.00 |
| | GRE+ | **38.41**±1.17 | **10.59**±1.17 | 0.82 | 29.26±2.10 | 39.74±2.10 | 1.00 | **58.29**±2.35 | 37.71±2.35 | 1.00 | **63.25**±2.25 | **3.29**±2.25 | 1.00 |

## 4.2    Experimental Results in the Independent and Sequential Editing Setting

In many real-world scenarios, well-trained models often produce inaccurate predictions on unseen data. To evaluate the practical effectiveness of editors for independent editing (**RQ1**), we randomly choose nodes from the validation set that were misclassified during the training. The editor is then

---

[3]MEND [11] and SERAC [12] are tailed for NLP application and are hard to extend to the graph area. MEND requires caching the input to each weight. Unfortunately, for graph data, the model edits cannot be done in a mini-batch way since the inference still runs in whole-batch, i.e., MEND requires caching the whole graph embedding at each layer.

[4]The experimental results on batch editing are in Appendix D.4.

applied to rectify the model's predictions for these misclassified nodes, and we evaluate the drawdown and edit success rate on the test set.

We edit one random single node 50 times and report the mean and standard deviation results in Tables 1 and 2 for small-scale and large-scale graph datasets, respectively. Our observations are made below:

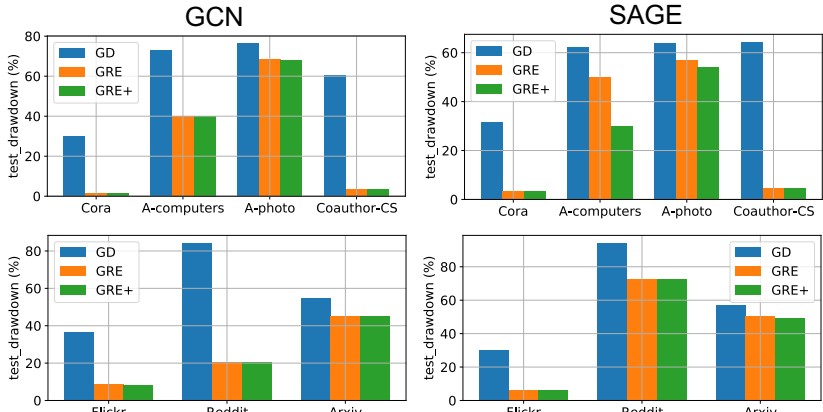

Figure 2: The test accuracy drawdown in sequential editing setting for GCN and GraphSAGE on various datasets. The units for y-axis are percentages (%).

❶ *Contrasting model editing on textual data [11, 12, 30], all editors can effectively rectify model predictions in the graph domain.* As shown in Table 1, all editors achieve a high success rate (typically from $96\% \sim 100\%$) after editing GNNs, which is highly different from transformers with below $50\%$ SR. This finding indicates that GNNs, unlike transformers, can be more easily adjusted to produce correct predictions. However, this improvement comes at the expense of **substantial drawdown** on other unrelated nodes, underscoring the key challenge of maintaining prediction locality for unrelated nodes before and after editing.

❷ *Our proposed GRE and GRE+ notably surpass both GD and ENN in terms of test drawdown.* This advantage stems mainly from the rewired gradient based on the pre-stored training loss gradient, which facilitates target sample correction while preserving the training loss. GD and ENN attempt to rectify model predictions by updating the parameters of GNNs without incorporating training loss information. In contrast, GRE and GRE+ maintain much better test accuracy after model editing. For example, for Amazon-photos, the accuracy drop dwindles from roughly $65.08\%$ to around $43.87\%$, a $43.9\%$ improvement over the baseline. This is due to the gradient rewiring approach that facilitates target sample correction while preserving the training loss. Interestingly, when applied to GNNs, ENN performs markedly worse than the basic editor GD. Moreover, GD performs well in MLP, which is consistent with the low gradient discrepancy of MLP.

❸ *Our proposed GRE and GRE+ are compatible with EGNN and further improve the performance.* We observe that while GRE occasionally underperforms, GRE+ consistently shows better performance than GD in reducing accuracy. For instance, when the A-computers dataset is evaluated with EGNN-GCN, GRE, and GRE+ exhibit an average accuracy drop of $4.62\%$ and $0.51\%$, respectively, whereas GD shows a decrease of $0.73\%$. Notably, we find that for 7 out of 8 datasets, GRE+ with EGNN-SAGE shows a negative drop in accuracy, meaning that the test accuracy actually increases after model editing. This points towards the superior performance of the EGNN-SAGE model architecture.

In the **sequential editing setting**, we select a sequence of nodes from the validation set that were misclassified during the training phase. The editor is then used to iteratively correct the model's predictions for these sequentially misclassified nodes, and we measure the resulting drawdown and success rate of edits on the test set.

In Figure 2, we report the test accuracy drawdown in the sequential setting, a more challenging scenario that warrants further investigation. In particular, we plot the test accuracy drawdown compared to GD across various GNN architectures and graph datasets. Our observations are as follows: ❹ *The proposed GRE and GRE+ consistently outperform GD in the sequential setting.* However, the drawdown is significantly higher than in the single edit setting. For instance, GRE+ exhibits a $43.87\%$ drawdown for GCN on the A-photo dataset in the single edit setting, which

escalates up to a $65\%$ drawdown in the sequential edit setting. These results also highlight the challenge of maintaining the locality of GNN prediction in sequential editing. ❺ *The improvement of GRE+ over GRE is quite limited in the sequential setting.* For example, GRE+ exhibits a $24.52\%$ drawdown over GRE for GCN on the A-photo dataset in the single edit setting while is on par with GRE in the sequential edit setting. These results further verify the difficulty of sequential editing and indicate more comprehensive training subset selection may be promising.

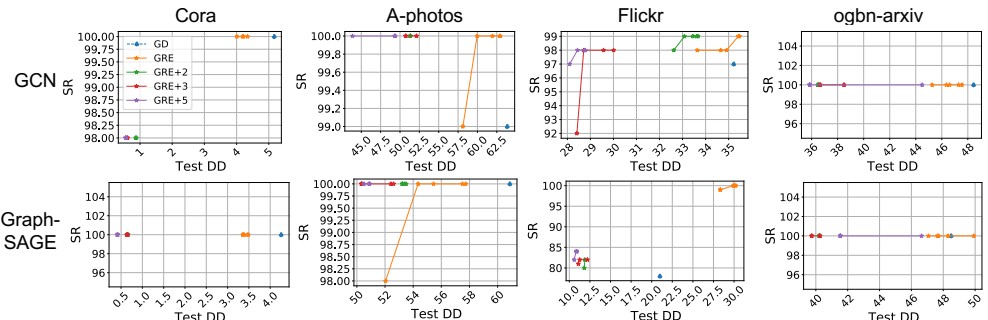

Figure 3: The success rate and test accuracy drawdown tradeoff in independent editing setting for GCN and GraphSAGE on various datasets. The trade-off curve close to the top left corner means better trade-off performance. The units for x- and y-axis are percentages (%).

## 4.3 Trade-off Performance Comparison

We further compare the trade-off between the accuracy drawdown and the success rate of our method on various GNN architectures and graph datasets. As shown in Figure 3, we plot Pareto front curves by assigning different hyperparameters for the proposed methods. The upper-left corner point represents the ideal performance, i.e., the highest SR and lowest accuracy drawdown. The results show that GRE+ achieves better trade-off results compared to GRE, and all methods consistently maintain a high success rate on various GNN architectures and graph datasets.

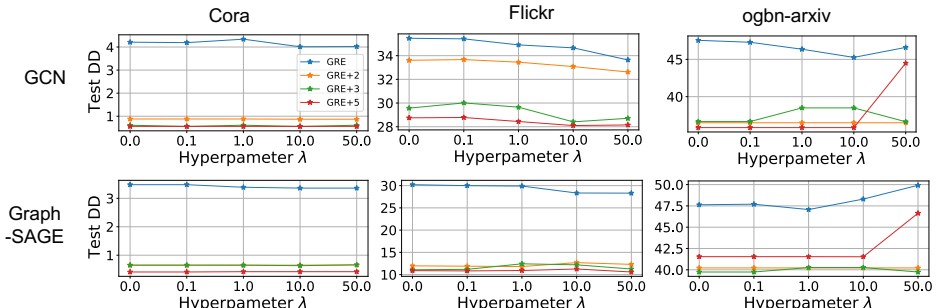

Figure 4: The hyperparameter study on test accuracy drawdown in independent editing setting w.r.t. $\lambda$.

## 4.4 Hyperparameter Study

In this experiment, we investigate the sensitivity of our proposed method w.r.t. $\lambda$ across a variety of GNN architectures and graph datasets. Specifically, we search for $\lambda$ from the set of $\{0.0, 0.1, 1.0, 10.0, 50.0\}$. As shown in Figure 4, the test accuracy drop remains relatively stable despite variations in $\lambda$, suggesting that meticulous tuning of this parameter may not be crucial. For the ogbn-arxiv dataset, an uptick in accuracy drop corresponds with an increase in $\lambda$, reflecting the inherent difficulty of this dataset. Intriguingly, in the case of GRE+5 with 5 training subsets, the test accuracy drop exceeds that of GRE+2 and GRE+3, a pattern that diverges from the trend observed in other datasets.

## 5   Conclusion

In this paper, we explore the editing of graph neural networks from a new gradient perspective. Through empirical observations, we discover that conventional model editing techniques often underperform due to the gradient discrepancy between the training loss and target loss in GNNs. To address this issue, we propose a gradient rewiring approach. Specifically, we formulate a constrained optimization problem to regulate the model performance during model editing and identify a simple yet effective gradient rewiring approach to explicitly satisfy the constraints. In this way, the proposed approach can correct the target sample while preventing an increase in training loss. Experiments demonstrate the effectiveness of our approach, and our proposed method is also compatible with the existing baseline EGNN and can further improve performance. Future work includes more comprehensive training subset selection in GRE+ and a tailed approach for editable graph neural networks training in the sequential editing setting.

## 6   Acknowledgements

The authors thank the anonymous reviewers for their helpful comments. This work is in part supported by NSF grants NSF IIS-2310260, IIS-2224843, IIS-2450662, IIS-2431515 and IIS-2239257. The views and conclusions contained in this paper are those of the authors and should not be interpreted as representing any funding agencies.

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

# A Experimental Setting

## A.1 More details on EGNN

We provide more details on editable graph neural networks (EGNN), a method free of neighborhood propagation, designed specifically for correcting misclassified node predictions [14]. EGNN uniquely integrates a peer MLP (matching in the number of hidden units and layers) with GNNs (such as GCN and GraphSAGE), and trains solely the MLP model using the target loss during model editing. This strategy enables EGNN to leverage the propagation-free advantages of MLPs for model editing. However, it's important to note that EGNN is incompatible with EGNN, as the model editing in EGNN refines a stitched MLP, which is not employed during model training. Our proposed methodologies, GRE and GRE+, are founded on gradient rewiring, which is orthogonal to the EGNN approach. In the appendix, we illustrate how our proposed methodologies can further augment the effectiveness of the EGNN approach and MLP models.

Table 3: Statistics information for datasets used for node classification.

| Datasets | Cora | A-computers | A-photo | Coauthor-CS | Flickr | Reddit | *ogbn-arxiv* | *ogbn-products* |
|---|---|---|---|---|---|---|---|---|
| # Nodes | 2,485 | 13,381 | 7,487 | 18,333 | 89,250 | 232,965 | 169,343 | 2,449,029 |
| # GREes | 5,069 | 245,778 | 119, | 81,894 | 899,756 | 23,213,838 | 1,166,243 | 61,859,140 |
| # Classes | 7 | 10 | 8 | 15 | 7 | 41 | 40 | 47 |
| # Feat | 1433 | 767 | 745 | 6805 | 500 | 602 | 128 | 218 |

## A.2 Datasets

The statistical information of all datasets is summarized in Table 3. The details of the datasets utilized for node classification are described as follows:

- **Cora** [31]: This citation network comprises 2,708 publications interconnected by 5,429 links. Each publication is characterized by a 1,433-dimensional binary vector that signifies the presence or absence of specific words from a predetermined vocabulary.

- **A-computers** [29]: This dataset is a segment of the Amazon co-purchase graph. In this network, nodes denote goods, and GREes represent frequent co-purchases of two goods. Node features are encoded as bag-of-words product reviews.

- **A-photo** [29]: Similar to A-computers, this is another segment of the Amazon co-purchase graph. Node features are also bag-of-words encoded product reviews.

- **Coauthor-CS** [29]: Derived from the Microsoft Academic Graph from the KDD Cup 2016 challenge 3, this co-authorship graph has nodes representing authors who are linked if they have co-authored a paper. Node features denote paper keywords for each author's publications, while class labels indicate an author's most active fields of study.

- **Reddit** [25]: This dataset is formulated from Reddit posts, with each node representing a post associated with different communities.

- **ogbn-arxiv** [3]: This dataset represents the citation network among all arXiv papers. Each node denotes a paper, and each GREe signifies a citation between two papers. Node features are generated from the average 128-dimensional word vector of each paper's title and abstract.

- **ogbn-products** [3]: This is an Amazon product co-purchasing network, where nodes represent Amazon products and GREes denote co-purchases of two products. Node features are created from low-dimensional representations of product description text.

## A.3 Implementation Details

The hyperparameters for model architecture, learning rate, dropout rate, and training epochs are shown in Table 4. For EGNN, we also adopt GNNs and MLPs with hyperparameters in Table 4. For GRE, we use the hyperparameters $\gamma = \{0.0, 0.1, 1.0, 10.0, 50.0\}$. For GRE+, we also select hyperparameters $\gamma = \{0.0, 0.1, 1.0, 10.0, 50.0\}$ and $K = \{1, 2, 3, 5\}$. As for QP problem Eq. (11), we use a standard package qpsolvers with version 3.4.0 to tackle this QP problem with ecos solver.

Table 4: Training hyperparameters configurations in the experiments

| Model | Configuration | Cora | A-computers | A-photo | Coauthor-CS | Flickr | Reddit | ogbn-arxiv | ogbn-products |
|---|---|---|---|---|---|---|---|---|---|
| **Graph-SAGE** | #Layers | 2 | 2 | 2 | 2 | 2 | 2 | 3 | 3 |
| | #Hidden | 32 | 32 | 32 | 32 | 256 | 256 | 128 | 256 |
| | lr | 0.01 | 0.01 | 0.01 | 0.01 | 0.01 | 0.01 | 0.01 | 0.002 |
| | Dropout | 0.1 | 0.1 | 0.1 | 0.1 | 0.3 | 0.5 | 0.5 | 0.5 |
| | Epoch | 200 | 400 | 400 | 400 | 400 | 400 | 500 | 500 |
| **GCN** | #Layers | 2 | 2 | 2 | 4 | 2 | 2 | 3 | 3 |
| | #Hidden | 32 | 32 | 32 | 32 | 256 | 256 | 128 | 256 |
| | lr | 0.01 | 0.01 | 0.01 | 0.01 | 0.01 | 0.01 | 0.01 | 0.002 |
| | Dropout | 0.1 | 0.1 | 0.1 | 0.1 | 0.3 | 0.5 | 0.5 | 0.5 |
| | Epoch | 200 | 400 | 400 | 400 | 400 | 400 | 500 | 500 |
| **MLP** | #Layers | 2 | 2 | 2 | 4 | 2 | 2 | 3 | 3 |
| | #Hidden | 32 | 32 | 32 | 32 | 256 | 256 | 128 | 256 |
| | lr | 0.01 | 0.01 | 0.01 | 0.01 | 0.01 | 0.01 | 0.01 | 0.002 |
| | Dropout | 0.1 | 0.1 | 0.1 | 0.1 | 0.3 | 0.5 | 0.5 | 0.5 |
| | Epoch | 200 | 400 | 400 | 400 | 400 | 400 | 500 | 500 |

## A.4   Running Environment

For hardware configuration, all experiments are executed on a server with 251GB main memory, 24 AMD EPYC 7282 16-core processor CPUs, and a single NVIDIA GeForce-RTX 3090 (24GB). For software configuration, we use CUDA=11.3.1, python=3.8.0, pytorch=1.12.1, higher=0.2.1, torch-geometric=1.7.2, torch-sparse=0.6.16 in the software environment. Additionally, we use the package of higher in https://github.com/eric-mitchell/mend for ENN implementation.

## B   Limitations and Discussions

While our proposed GRE and GRE+ methods effectively mitigate the accuracy dropdown compared to conventional gradient descent algorithms, the success of our approaches is largely contingent on the precision of the pre-stored gradient for training loss. Despite the relatively few required model edit steps for single node editing, the accuracy of the pre-stored gradient may not sustain over long-term model editing, as the pre-stored gradient for training loss could exhibit significant discrepancy from the gradient of training loss for the edited model. To address such discrepancy, a straightforward strategy could involve leveraging critical training samples to estimate the true gradient of training loss for the edited model. Another possible direction is to identify critical samples instead of random samples for GRE+ with the aim of further constraining the model's behavior before and after model editing.

Notice that the proposed gradient rewiring method is not inherently specific to graphs, the gradient rewiring method is particularly suitable in the graph domain due to the small model size. Specifically, graph models are typically a few layers and thus are smaller in model size compared to models (e.g., Transformers) used in NLP and CV tasks. This results in lower computational and storage costs for gradients, making our strategy particularly suitable for the graph domain. Additionally, it is more challenging to edit nodes in a graph due to the inherent propagation process within neighborhoods. Such propagation may lead to significant gradient discrepancies within the graph domain.

## C   Algorithms

We show the algorithms of GRE and GRE+ during model editing in Algorithm 1 and 2, respectively.

## D   More Experimental Results

In this section, we present experimental results to showcase the improved efficacy of our proposed methods, GRE and GRE+. These techniques enhance the performance of EGNN, a specifically designed editable graph neural network, across both independent and sequential editing settings.

**Algorithm 1** Gradient Rewiring Editable (GRE) Graph Neural Networks Training

1: **Input:** Target samples $(\mathbf{x}_{tg}, \mathbf{y}_{tg})$, hyperparameter $\lambda$, well-trained GNN model $f_\theta(\cdot)$, and its corresponding gradient for the training subgraph.
2: **Output:** Updated GNN model $f_{\theta'}(\cdot)$.
3: **while** $f_\theta(\mathbf{x}_{tg}) \neq \mathbf{y}_{tg}$ **do**
4:     Compute the model gradient $g_{tg}$ for the target loss $\mathcal{L}_{tg}$.
5:     Rewire the target loss gradient $g_{tg}$ by reducing the projection component on $g_{train}$, then scale with $(1 + \lambda)^{-1}$:
6:         $g^* = (1 + \lambda)^{-1} (g_{tg} - v^* g_{train})$.
7:     Replace $g_{tg}$ with $g^*$ and update the model parameters using the optimizer to obtain $\theta'$.
8: **end while**

---

**Algorithm 2** Gradient Rewiring Editable Plus (GRE+) Graph Neural Networks Training

1: **Input:** Target samples $(\mathbf{x}_{tg}, \mathbf{y}_{tg})$, hyperparameters $\lambda$, well-trained GNNs model $f_\theta(\cdot)$, and its corresponding model gradient for training subgraph.
2: **Output:** Editable GNNs model $f_{\theta'}(\cdot)$.
3: **while** $f_\theta(\mathbf{x}_{tg})! = \mathbf{y}_{tg}$ **do**
4:     Compute model gradient $g_{tg}$ for target loss $\mathcal{L}_{tg}$.
5:     Solve QP problem Eq. (11) via standard QP solver package and obtain the optimal dual variable $\mathbf{v}^*$.
6:     Calculate the rewired gradient using $g^* = (1 + \lambda)^{-1}(\boldsymbol{G}\mathbf{v}^* + g_{tg})$.
7:     Replace $g_{tg}$ with $g^*$ and then adopt optimizer to update model parameters as $\theta'$.
8: **end while**

## D.1 More results on Model Architectures for Independent Editing

In this section, we conduct a sequence of 50 single-node edits and present the mean and standard deviation results in Tables 5 for large-scale graph datasets. Similarly, GRE occasionally underperforms, and GRE+ consistently shows better performance than GD with respect to the reduction in accuracy. For instance, when the Reddit dataset is evaluated with EGNN-GCN, GRE and GRE+ exhibit an average accuracy drop of $1.48\%$ and $-0.21\%$, respectively, whereas GD shows a decrease of $1.28\%$. Moreover, GRE+ with EGNN-SAGE shows a negative drop in accuracy among 6 out of 8 datasets, i.e., the test accuracy actually increases after model editing.

Table 5: The results on four large scale datasets after applying one single edit. "OOM" is the out-of-memory error.

| | Editor | Flickr | | | Reddit | | | ogbn-arxiv | | | ogbn-products | | |
|---|---|---|---|---|---|---|---|---|---|---|---|---|---|
| | | Acc↑ | DD↓ | SR↑ | Acc↑ | DD↓ | SR↑ | Acc↑ | DD↓ | SR↑ | Acc↑ | DD↓ | SR↑ |
| MLP | GD | 35.84±1.23 | 11.16±1.23 | 0.83 | **45.27±0.97** | **7.73±0.97** | 0.99 | **70.31±0.40** | **2.69±0.40** | 1.00 | **74.19±3.40** | **0.20±3.40** | 1.00 |
| | ENN | 25.82±14.90 | 25.38±16.90 | 1.00 | 11.16±5.10 | 84.36±3.10 | 1.00 | 16.59±7.70 | 53.62±6.70 | 1.00 | OOM | OOM | 0 |
| | GRE | 36.47±0.57 | 10.53±0.57 | 0.81 | 35.85±2.60 | 17.15±2.60 | 1.00 | 62.20±0.94 | 10.80±0.94 | 1.00 | 53.99±0.60 | 20.01±0.60 | 1.00 |
| | GRE+ | **43.23±0.17** | **3.77±0.17** | 0.84 | 41.33±0.87 | 11.67±0.87 | 0.99 | 64.11±0.95 | 8.40±0.95 | 1.00 | 57.43±1.30 | 16.89±1.30 | 1.00 |
| EGNN-GCN | GD | 46.10±0.91 | 4.90±0.91 | 0.93 | 68.72±0.55 | 1.28±0.55 | 1.00 | 86.08±0.83 | -0.08±0.17 | 1.00 | 73.73±0.12 | 0.27±0.12 | 1.00 |
| | GRE | 45.70±0.97 | 5.30±0.97 | 0.94 | 68.52±0.51 | 1.48±0.51 | 1.00 | **89.22±0.34** | **−3.22±1.34** | 1.00 | 73.65±0.16 | 0.35±0.16 | 1.00 |
| | GRE+ | **50.60±0.15** | **0.40±0.15** | 0.99 | **69.97±0.38** | **−0.21±0.38** | 1.00 | 88.51±0.57 | −2.51±0.43 | 1.00 | **74.06±0.45** | **−0.80±0.45** | 1.00 |
| EGNN-SAGE | GD | 45.68±1.15 | 2.32±1.15 | 0.95 | 67.76±0.53 | 1.24±0.53 | 1.00 | 95.99±0.02 | 0.01±0.02 | 0.98 | 75.89±0.06 | 0.11±0.06 | 1.00 |
| | GRE | 42.25±1.64 | 5.75±1.64 | 1.00 | 67.34±0.35 | 1.66±0.35 | 0.99 | 94.09±1.29 | 1.91±1.29 | 1.00 | 75.90±0.05 | 0.10±0.05 | 1.00 |
| | GRE+ | **49.06±1.42** | **−1.05±1.42** | 1.00 | **68.48±0.78** | **0.11±0.78** | 1.00 | **96.06±0.10** | **−0.06±0.10** | 0.95 | **76.26±0.21** | **−0.17±0.21** | 1.00 |

## D.2 Experimental Results on other Model Architectures for Sequential Editing Setting

In the sequential editing setting, we take a sequence of 50 misclassified nodes and use the editor to iteratively correct the model's predictions for EGNN across different GNN architectures. The test accuracy drop associated with various model editing methods for different graph datasets is reported in Figure 7. Our observations indicate that the proposed GRE and GRE+ methods consistently outshine GD in this sequential setting. For instance, with the Coauthor-CS dataset and EGNN-GCN, our proposed methods achieve virtually no decrease in accuracy, while GD exhibits a drop of over $7\%$. Another compelling observation is that the improvement demonstrated by our methods over GD for EGNN is markedly larger than for GNNs. This suggests potential synergies between optimizer selection and model architecture design.

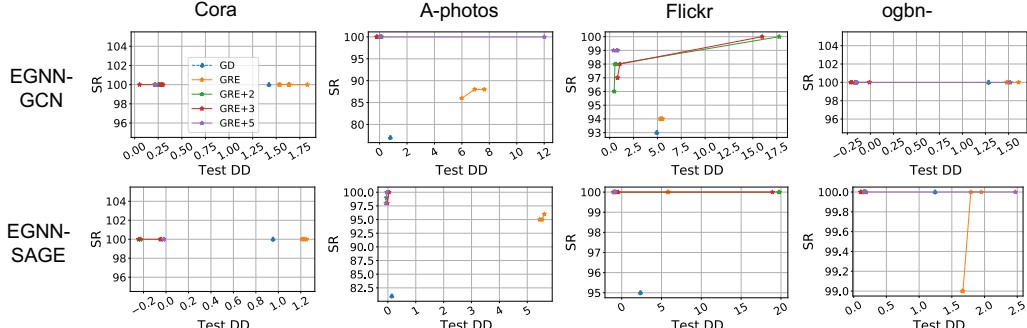

Figure 5: The success rate and test accuracy dropdown tradeoff in independent editing setting for EGNN-GCN and EGNN-SAGE on various datasets. The trade-off curve close to the top left corner means better trade-off performance. The units for the x- and y-axis are percentages (%).

## D.3  Trade-off Performance Comparison on other Model Architectures

We extend our evaluation by comparing the trade-off between the accuracy drop and success rate of our method on EGNN across various graph datasets. By adjusting different hyperparameters for the proposed methods, we construct Pareto front curves as shown in Figure 5. The results underscore that both GRE+ and GRE outperform GD in achieving superior trade-off outcomes. Importantly, our proposed methods exhibit robust preservation of the success rate across various GNN architectures and graph datasets.

## D.4  More Experimental results on Batch Editing

In this section, we present the experimental results of applying batch editing on four small-scale datasets (Table 6) and four large-scale datasets (Table 7).

For the small-scale datasets, our proposed methods, GRE and GRE+, consistently outperform the baseline methods. For example, on the Cora dataset, GRE+ achieves the highest accuracy with a minimal drawdown and a high success rate. Specifically, GRE+ can reduce 56.9% and 30.3% drawdown compared with 2nd best baseline in GCN and GraphSAGE architectures, respectively. For the large-scale datasets, GRE+ again demonstrates superior performance. On ogbn-products datasets GRE+ can reduce 2.5% and 0.5% drawdown compared with 2nd best baseline GRE in GCN and GraphSAGE architectures, respectively, while maintaining a high success rate.

Table 6: The results on four small-scale datasets after applying batch edit. **SR** is the edit success rate, **Acc** is the test accuracy after editing, and **DD** are the test drawdown, respectively. The best/second-best results are highlighted in **boldface**/underlined, respectively.

| | Editor | Cora | | | A-computers | | | A-photo | | | Coauthor-CS | | |
|---|---|---|---|---|---|---|---|---|---|---|---|---|---|
| | | Acc↑ | DD↓ | SR↑ | Acc↑ | DD↓ | SR↑ | Acc↑ | DD↓ | SR↑ | Acc↑ | DD↓ | SR↑ |
| GCN | GD | 81.72±5.24 | 7.28±4.04 | 0.77 | 60.98±22.41 | 27.02±3.89 | 0.53 | 44.32±11.26 | 49.64±11.54 | 0.52 | 67.64±6.23 | 26.36±7.05 | 1.00 |
| | ENN | 32.16±2.75 | 46.12±1.84 | 0.93 | 25.91±13.01 | 27.09±14.51 | 0.25 | 9.99±0.81 | −0.99±0.99 | 0.06 | 45.59±13.21 | −43.59±15.54 | 0.62 |
| | GRE | 82.96±2.47 | 6.04±3.29 | 0.96 | 64.67±3.47 | 23.33±13.85 | 0.63 | 54.66±26.26 | 39.34±28.82 | 0.34 | 76.24±5.29 | 17.76±5.87 | 1.00 |
| | GRE+ | **86.40±0.76** | **2.60±0.55** | 0.97 | **65.25±0.35** | **22.70±0.65** | 0.62 | **57.83±1.36** | **36.13±9.25** | 0.50 | **76.80±9.56** | 17.20±9.93 | 1.00 |
| Graph-SAGE | GD | 78.48±4.33 | 8.52±1.70 | 1.00 | 29.95±18.28 | 53.08±9.55 | 0.53 | 46.98±15.24 | 47.02±17.52 | 0.50 | 67.64±7.58 | 23.27±7.97 | 1.00 |
| | ENN | 32.16±2.21 | 45.88±1.68 | 0.99 | 0.99±0.00 | 0.08±0.01 | 0.08 | 14.81±7.92 | −4.81±18.23 | 0.17 | **77.55±2.12** | −15.55±2.07 | 1.00 |
| | GRE | 80.68±1.17 | 6.32±1.17 | 0.99 | 46.58±2.25 | 36.42±10.43 | 0.54 | 56.95±18.27 | 37.05±20.18 | 0.46 | 75.68±8.44 | 19.32±8.99 | 1.00 |
| | GRE+ | **82.60±0.87** | **4.40±1.07** | 1.00 | **51.24±12.87** | **32.51±15.77** | 0.62 | **62.60±11.82** | **31.40±12.93** | 0.52 | 76.51±6.21 | 18.49±6.80 | 1.00 |

Table 7: The results on four large-scale datasets after applying batch edit. "OOM" is the out-of-memory error. The best/second-best results are highlighted in **boldface**/underlined, respectively.

| | Editor | Flickr | | | Reddit | | | ogbn-arxiv | | | ogbn-products | | |
|---|---|---|---|---|---|---|---|---|---|---|---|---|---|
| | | Acc↑ | DD↓ | SR↑ | Acc↑ | DD↓ | SR↑ | Acc↑ | DD↓ | SR↑ | Acc↑ | DD↓ | SR↑ |
| GCN | GD | 19.79±12.12 | 31.21±12.55 | 0.29 | 37.64±5.30 | 58.36±5.20 | 1.00 | 38.61±4.91 | 31.39±5.57 | 0.83 | 41.83±5.94 | 32.17±4.50 | 1.00 |
| | ENN | **42.82±1.92** | **0.00±1.90** | 0.00 | **65.70±3.11** | **−59.70±3.24** | 1.00 | 24.47±1.77 | -18.41±1.81 | 0.77 | OOM | OOM | 0 |
| | GRE | 24.90±5.51 | 26.10±5.31 | 0.39 | 24.74±1.92 | 45.26±1.92 | 1.00 | 41.96±7.26 | 29.04±7.51 | 0.62 | 42.52±4.60 | 31.48±4.86 | 1.00 |
| | GRE+ | 25.10±6.67 | 25.90±6.47 | 0.42 | 52.61±4.23 | 43.59±5.81 | 1.00 | **41.13±4.10** | **28.87±5.04** | 0.80 | **42.60±4.89** | **31.40±5.03** | 1.00 |
| Graph-SAGE | GD | 20.71±11.20 | 18.29±10.05 | 0.27 | 29.65±22.5 | 66.35±5.20 | 1.00 | 41.05±6.81 | 27.95±7.87 | 0.77 | 49.33±4.18 | 17.15±5.21 | 0.94 |
| | ENN | **41.89±0.60** | **−16.89±0.03** | 0.56 | 17.10±4.90 | -15.10±5.56 | 0.24 | 13.82±2.68 | 8.18±6.52 | 0.26 | OOM | OOM | 0 |
| | GRE | 26.92±2.62 | 22.08±3.06 | 0.59 | 31.40±8.94 | 64.60±9.76 | 0.93 | 38.65±7.22 | 30.35±8.84 | 0.70 | 50.21±3.82 | 16.33±4.92 | 0.92 |
| | GRE+ | 27.97±1.17 | 21.03±7.97 | 0.42 | 38.01±7.32 | **56.99±6.88** | 0.95 | **42.76±4.31** | **26.24±8.47** | 0.79 | **50.30±5.83** | **16.24±6.25** | 0.90 |

## D.5 The Edit Time and Memory Comparison for Editing Methods

In this section, we present the experimental results of the edit time and memory required for editing across four large-scale datasets (Table 8).

We observe that GRE+ takes $1.5\ 2.5\times$ wall-clock editing time than the GD/GRE editor in terms of the wall-clock edit time. This is because GRE+ requires QP solver to obtain the rewired gradient. In terms of memory consumption, the overall memory overhead is insignificant. For example, GRE+ (5) requires $17.9\%$ GPU memory than GD editor in obgn-products dataset and GCN architecture. The reason is that the anchor gradient is required to store in memory and QP solver computation in memory.

Table 8: The edit time and memory required for editing. ET (ms) and PM (MB) represent the edit time in milliseconds and peak memory in megabytes, respectively.

|  | Editor | Flickr | | Reddit | | ogbn-arxiv | | ogbn-products | |
|---|---|---|---|---|---|---|---|---|---|
|  |  | ET (ms) | PM (MB) | ET (ms) | PM (MB) | ET (ms) | PM (MB) | ET (ms) | PM (MB) |
| GCN | GD | 67.46 | 707.0 | 345.23 | 3244.8 | 94.58 | 786.2 | 2374.15 | 14701.7 |
|  | ENN | 109.82 | 666.8 | 405.24 | 3244.8 | 242.85 | 786.2 | – | OOM |
|  | GRE | 63.93 | 695.8 | 391.54 | 3491.3 | 84.74 | 956.9 | 2400.78 | 17336.6 |
|  | GRE+ (2) | 100.45 | 696.0 | 457.08 | 3493.2 | 121.11 | 957.8 | 2413.69 | 17338.7 |
|  | GRE+ (3) | 115.29 | 697.9 | 509.44 | 3493.9 | 131.06 | 957.9 | 2471.23 | 17338.9 |
|  | GRE+ (5) | 155.05 | 698.6 | 603.85 | 3495.6 | 162.24 | 958.3 | 2591.06 | 17339.2 |
| Graph-SAGE | GD | 117.74 | 843.0 | 1024.12 | 4416.53 | 107.63 | 891.3 | 2125.07 | 13832.2 |
|  | ENN | 134.50 | 843.0 | 2597.21 | 4416.5 | 277.29 | 891.3 | – | OOM |
|  | GRE | 116.03 | 952.4 | 1089.29 | 4955.4 | 100.09 | 1072.5 | 2132.02 | 16254.1 |
|  | GRE+ (2) | 167.17 | 954.5 | 1267.13 | 4959.0 | 136.28 | 1073.7 | 2135.88 | 16255.9 |
|  | GRE+ (3) | 176.66 | 955.5 | 1363.53 | 4960.7 | 154.29 | 1074.0 | 2211.63 | 16256.0 |
|  | GRE+ (5) | 219.81 | 957.5 | 1603.03 | 4964.2 | 180.73 | 1075.5 | 2275.72 | 16256.3 |

## D.6 More Test Accuracy Results on Sequential Editing

In this subsection, we present the after-editing test accuracy results of applying sequential editing on various datasets for both GCN and GraphSAGE models in Figure 6. The test accuracy is reported as a percentage for each dataset.

Overall, our proposed methods, GRE and GRE+, consistently outperform the baseline methods GD and ENN across all datasets in terms of test accuracy. For example, on the Reddit dataset, the proposed methods can achieve more than $100\%$ improvement over GD and ENN in terms of accuracy. Besides, compared with GRE and GRE+, the improvement is marginal on most occasions except A-computer dataset in GraphSAGE, which indicates the limited effectiveness of the fine-grained gradient rewiring in GRE+.

# E  More Related Work

**Gradient-based method for other tasks.**   The existing literature on gradient modification mainly incorporates continual learning and meta learning. In continual learning, work [32] proposes gradient projection methods to update the model with gradients in the orthogonal directions of old tasks, without access to old task data. GPM [33] identifies the bases of these subspaces by examining network representations after learning each task using Singular Value Decomposition (SVD) in a single-shot manner and stores them in memory as gradient projection memory. Class gradient projection is proposed in [34] to address the class deviation in gradient projection. In meta-learning, work [35] proposes a meta-learning algorithm to learn to modulate the gradient in the absence of abundant data. The implicit model-agnostic meta-learning (iMAML) algorithm is developed in [36] for optimization-based meta-learning with deep neural networks that remove the need for differentiating through the optimization path. [37] provides a theoretical framework for designing and understanding practical meta-learning methods that integrate sophisticated formalizations of task-similarity.

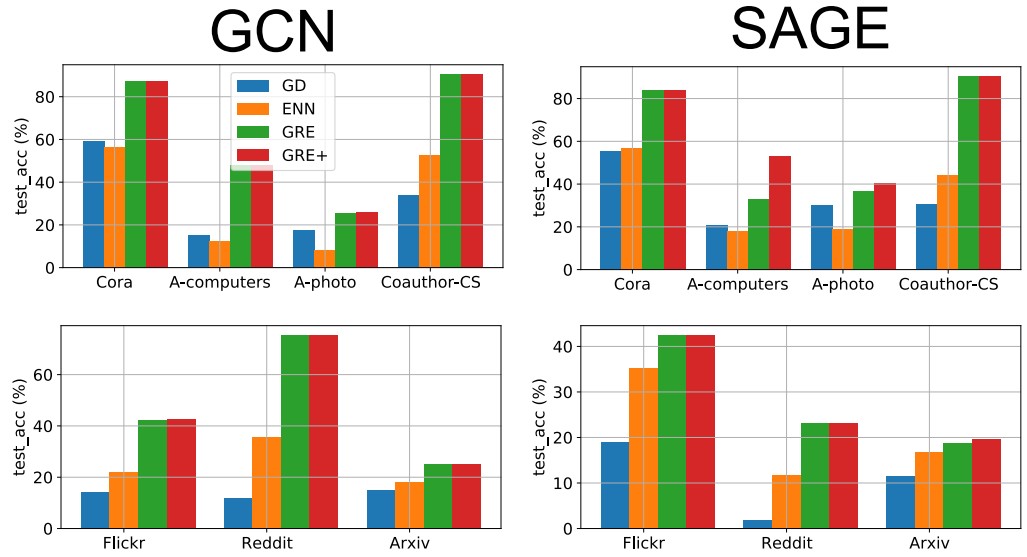

Figure 6: The test accuracy in sequential editing setting for GCN and GraphSAGE on various datasets. The units for y-axis are percentages (%).

## F    More discussion

**Comparison with Curriculum Learning.**    Curriculum learning and model editing are two distinct approaches in the field of machine learning. Curriculum learning is an approach where the network is trained in a structured manner, starting with simpler tasks and gradually introducing more complex ones. This method aims to improve the learning process by mimicking how humans learn. Model editing is a fast and efficient approach to patch the well-trained model prediction for several failed test cases. Although both are multi-stage training stages, there are several key differences: (1) Goals: Curriculum Learning aims to improve the overall learning process by structuring the training data in a way that mimics human learning. In contrast, model editing aims to make targeted adjustments to a pre-trained model to correct undesirable behaviors. (2) Approach: curriculum learning mainly focuses on the sequence and complexity of the training data. Model editing typically modifies the model's parameters or architecture to correct undesirable behavior goals. (3) Additional information in the multi-stage process. Model editing requires failure feedback for well-trained models as the target samples to patch, e.g., test failure cases after production is launched. In other words, such feedback can only be obtained after model pertaining. In curriculum learning, all information is given in multi-stage training. In summary, curriculum learning focuses on structuring the training process to improve overall learning, while model editing focuses on making targeted adjustments to a pre-trained model to correct specific behaviors. Both approaches can be complementary and used together to achieve better model performance.

**Comparison with Domain Adaptation.**    To the best of our knowledge, many existing methods in domain adaptation (DA) [38, 39] integrate source and target gradients in the loss function. For example,  [38] aims to minimize the gradient discrepancy for unsupervised DA, and  [39] aligns gradient distribution for better adversarial DA. However, these methods can not be applied in graph model editing since (1) gradient discrepancy is required to successfully edit model prediction; (2) model editing collapses (i.e., no gradient discrepancy) at the initial stage; (3) regulating gradient behavior is insufficient for model editing task since the main problem is how to get an edited model instead of cross-domain generalization. To this end, we rewire the gradient before the model parameters update, and derive a closed-form, instead of a learning-based, gradient rewiring method to accelerate model editing.

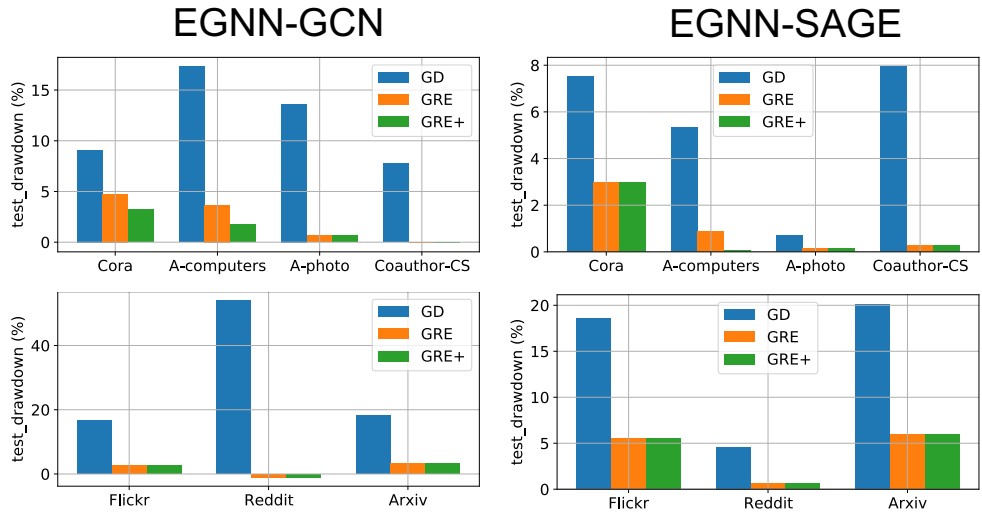

Figure 7: The test accuracy dropdown in sequential editing setting for EGNN-GCN and EGNN-SAGE on various datasets. The units for the y-axis are percentages (%).

