# OpenReview forum: "Gradient Rewiring for Editable Graph Neural Network Training"
_NeurIPS.cc/2024/Conference — NeurIPS 2024 poster_

### Official Review · Reviewer_X56x · 2024-06-18

**Soundness:** 3
**Presentation:** 2
**Contribution:** 3
**Rating:** 5
**Confidence:** 4

**Summary:**

Model editing involved fine-tuning a pre-trained model specifically on training examples where it cannot predict the correct output in order to correct these errors. While model training has been explored for CV and NLP models, GNNs pose a unique challenge due to the unordered data type and node-level classification problem. This paper attempts to rectify this error by proposing a Gradient Rewiring+ (GRE+) method for Editable graph neural network training while preserving the original train/test set performance. GRE+ is evaluated on several small and large classification datasets.

**Strengths:**

- The proposed method aims to rectify errors that still persist after training, without compromising the original performance. To quantify this, the authors report three metrics: The accuracy after model editing has been performed, the DropDown metric measuring how this impacts the original performance and the success rate of the method at actually rectifying the error. The three of these provide a clear, multi-faceted view of the effect of the proposed method against the baselines.
- In most situations, the proposed GRE+ proves to be the best method.
- The introduction and motivation of the paper is generally well-written and easy to follow.

**Weaknesses:**

- Biggest difference is the presentation and decision to treat GRE and GRE+ as different methods to compare to baseline approaches. This is because the experimental results in Tables 1 and 2 show that GRE on its own is not really noteworthy as it fails to consistently rank as the 2nd best behind GRE+ and is outperformed by GD and ENN on numerous occassions.
- This hampers the writing in the method which focuses on what GRE does first, before expanding to talk about GRE+ with more specifics. In contrast to this writing decision, it would be better if GRE and GRE+ were not treated separately, GRE meant GRE+ and instead of having 2 rows for your method in Tabs 1/2, you just had GRE(+), then later did an ablation study (where one option is the GRE config as mentioned in the submitted paper) to show the efficacy/necessity of the whole method.
- Line 243 This is the first time the word "Transformers" is mentioned in the paper and its jarringly brought up out of the blue. The introduction broadly mentioned that model editing exists for CNNs and NLP tasks but not specifically the Transformer architecture. This prose should be deeply revised before the paper is publication ready.
- There is no analysis/study of the cost/complexity of the model editing methods which is a major weakness. Looking at Fig 3 while it is true that GRE+ improves little/none over GRE, in Tab 1/2 the story is much different. There does not seem to be a situation where GRE is better so I must question why it is presented as a distinct approach itself compared to GRE+. Is it because it is computationally less expensive?
- Some statements are not properly substantiated with citations:
	- L105 "It is well-known that model editing incurs training performance degredation"
	- L167 "Since shallow GNNs model performs well in practice" seems like a handwave. Also, spelling/grammar check.
- Other nitpicks:
	- Tab 1 caption mentions "OOM" for some methods but that is only used in entries for Tab - Fig 2 should have larger fonts it is very hard to read.
	- L242, should read ", respectively." when enumerating like that.

**Questions:**

- Fig 3 where are results on ENN?

**Limitations:**

A limitations section is provided in the supplementary material.

---

> ### Author Rebuttal · Authors · 2024-08-07
>
> Thanks for the constructive and insightful comments. We carefully revised the manuscripts based on all reviewers' comments. Please see the revised manuscript at https://anonymous.4open.science/r/Gradient_rewiring_editing-E16E/GRE_NeurIPS24.pdf.
>
> **[W1: Presentation on GRE and GRE+]**
>
> Ans: hank you for the constructive comment. We respectfully disagree with the assertion that GRE isn't consistently ranked as the second-best method behind GRE+. We believe that treating GRE and GRE+ as different methods in our writing is reasonable.
>
> GRE achieves second-best performance in **both sequential editing and batch editing**, while demonstrating competitive performance in independent editing. In the independent editing task, all methods can achieve a high success rate since it is relatively simple to correct one target sample. However, GRE still secures second-best performance in several instances, such as on the ogbn-arxiv and ogbn-products datasets in GCN. More importantly, GRE excels in more complex editing scenarios, such as sequential editing and batch editing. For sequential editing, GRE achieves second-best performance across various datasets, as shown in **Figures 3 and 6**. Similarly, for batch editing, GRE also ranks second-best, as evidenced by the results in **Tables 6 and 7**.
>
> In summary, GRE shows significant improvement over baselines in sequential and batch editing. From a technical perspective, GRE provides a closed-form solution for gradient rewiring, while GRE+ requires a numerical QP solver. Therefore, we treat GRE and GRE+ separately due to the significant performance improvements offered by GRE and its distinct closed-form solution.
>
>
>
> **[W2: Model editing exists for CNNs and NLP tasks but not specifically the Transformer architecture]**
>
> Ans: To avoid confusion, we have revised the statement of Observation 1 in Section 4.2. Specifically, we removed the part about transformers and generally mentioned that all editors can effectively rectify model predictions for independent editing in the graph domain.
>
> **[W3: No analysis/study of the cost/complexity of the model editing methods]**
>
> Ans: For comparing time complexity and memory consumption, we measured the wall-clock editing time in milliseconds and GPU peak memory in megabytes, reporting average values over 50 independent edits in **Appendix D.5**. Due to the limited rebuttal length, we only show time complexity and memory consumption for GCN.
>
> Our results show that the proposed method is scalable in terms of memory consumption and has manageable editing time overhead. For example, in the GraphSAGE architecture on the Flickr dataset, GRE+ (5) results in only a 13.5% increase in peak memory compared to GD. In terms of wall-clock editing time, the most time-consuming version, GRE+ (5), shows an insignificant overhead, with only a 6.31% increase on the ogbn-products dataset in the GraphSAGE architecture. These observations demonstrate the scalability of GRE+ for large datasets. It is noteworthy that model editing is usually efficient and fast, making slightly slower editing affordable while improving editing effectiveness.
>
> |                |       Editor        | Flickr ET (ms) | Flickr PM (MB) | Reddit ET (ms) | Reddit PM (MB) | ogbn-arxiv ET (ms) | ogbn-arxiv PM (MB) | ogbn-products ET (ms) | ogbn-products PM (MB) |
> |----------------|---------------------|----------------|----------------|----------------|----------------|--------------------|--------------------|-----------------------|-----------------------|
> | **GCN**        | **GD**              | 67.46          | 707.0          | 345.23         | 3244.8         | 94.58              | 786.2              | 2374.15               | 14701.7               |
> |                | **ENN**             | 109.82         | 666.8          | 405.24         | 3244.8         | 242.85             | 786.2              | --                    | OOM                   |
> |                | **GRE**             | 63.93          | 695.8          | 391.54         | 3491.3         | 84.74              | 956.9              | 2400.78               | 17336.6               |
> |                | **GRE+ (2)**        | 100.45         | 696.0          | 457.08         | 3493.2         | 121.11             | 957.8              | 2413.69               | 17338.7               |
> |                | **GRE+ (3)**        | 115.29         | 697.9          | 509.44         | 3493.9         | 131.06             | 957.9              | 2471.23               | 17338.9               |
> |                | **GRE+ (5)**        | 155.05         | 698.6          | 603.85         | 3495.6         | 162.24             | 958.3              | 2591.06               | 17339.2               |
>
>
> **[W4: Add citations and other nitpicks.]**
>
> Ans: We have added citations for several statements and revised the format and writing issues accordingly.
>
> **[Q1: Fig 3 where are results on ENN?]**
>
> Ans: Thank you for your constructive comment. Model editing consists of two stages: the pre-training stage, where a well-trained model is obtained, and the editing stage, where undesirable behavior is corrected. The GD, GRE, and GRE+ methods perform editing during the editing stage using the **same pre-trained model**. For these methods, a higher test accuracy drawdown implies a lower test accuracy after editing. In contrast, the key goal of the ENN baseline is to improve the model during the pre-training stage. The objective at this stage is to obtain a model that can be easily edited using gradient descent while maintaining good task performance. Since **the test accuracy of the pre-edited models varies**, it is infeasible to compare these methods in terms of test drawdown. To fairly compare all methods, we have **added a comparison of the test accuracy performance after editing for sequential editing in Appendix D.6**. It is observed that the accuracy of GRE+ is higher than all baselines.

---

> > ### Comment · Reviewer_X56x · 2024-08-09
> >
> > I thank the authors for their detailed rebuttal and paper revisions. While I am not totally convinced by the decision to differentiate GRE and GRE+, I do find the rebuttal rational is reasonable. After reading the other reviews and responses, I am willing to raise my score to borderline accept.

---

> > > ### Author Response · Authors · 2024-08-12
> > > **Response to Reviewer X56x**
> > >
> > > Thank you for your thoughtful review and for considering our detailed rebuttal and revisions. We appreciate your feedback on the differentiation between GRE and GRE+, and we're glad you found our rationale reasonable. We're grateful for your willingness to raise your score and value your insights in helping us refine our work.

---

### Official Review · Reviewer_KXu3 · 2024-06-25

**Soundness:** 3
**Presentation:** 2
**Contribution:** 3
**Rating:** 5
**Confidence:** 3

**Summary:**

The authors propose a model editing technique motivated by an observed inconsistency between the gradient of target and training nodes' cross-entropy losses. Their proposed method, Gradient Rewiring for Editable Graph Neural Networks (GRE), stores the anchor of the gradients of the training nodes and uses the anchors during editing to preserve performance. Finally, the authors empirically evaluate their proposed method on a collection of real-world datasets.

**Strengths:**

- I believe the problem to be an important practical problem and that the paper is well-motivated.
- The main observation of the paper, i.e., that the gradients of the target and the training nodes are inconsistent, is interesting and novel in the context of Graph Neural Networks.
- The proposed method seems to perform well consistently. The authors have performed a comprehensive empirical evaluation, and I believe the results are solid.
- The authors are overcoming the quadratic problem in the number of model parameters by solving a dual problem with significantly fewer variables.

**Weaknesses:**

- The method requires storing the gradients of the training dataset. This would raise the memory consumption. It would be great if the authors could include a table with the memory requirements for different models.
- There are a lot of writing/grammar issues. In the following, I will give a (non-exhaustive) list of examples of missing articles and text where the writing could generally be improved. This would generally not be a big deal, but in this case, the writing makes the text very hard to follow and comprehend. I would encourage the authors to do a few revision rounds before updating the manuscript. In my personal experience, free tools such as Grammarly also help with identifying weird formulations or grammar mistakes:
	- 16-17: the gradient of *the* loss
	- Line 21: "interpreting the features and topology of graph data" - *interpreting* is a somewhat unusual way to describe learning representations on graphs; please consider rephrasing.
	- Line 42: "(...) intricate editing for GNNs through the lens of landscape." - this phrasing is very confusing; what landscape? I suggest the authors rephrase to something more akin to "through the lens of the loss landscape of the Kullback-Lieber divergence between the initial node features and the final node embeddings" since this seems to be the technique that [[1]] is proposing.
	- Line 46: "(...) perspective, and is compatible with existing work" - please consider rephrasing to something similar to "(...) perspective, which is compatible with existing work".
	- Line 49: "(...) can lead to a deterioration in the performance *of* the training nodes" -> "(...) can lead to a deterioration in the performance *on* the training nodes" - the performance is of the model on the training nodes
	- Line 55: similar, "performance *of* the training nodes" -> "performance *on* the training nodes"
	- The caption of Figure 1 is somewhat confusing; please consider rephrasing.
	- Line 93-94: Somewhat confusing: "motivation to rewire gradients *of* model editing" or "motivation to rewire gradients *for* model editing"?
	- Line 95: "and advanced version (GRE+)" -> "and *an* advanced version (GRE+)"
	- Lines 98-99: "we pre-train (...) on (...) datasets" -> "we pre-train (...) on *the* (...) datasets"
	- Lines 101-102: please consider rephrasing to something more akin to "we fine-tune the well-trained model using the cross-entropy loss of the target sample via gradient descent"
	- Lines 106-108: "we investigate performance degradation from model gradient perspective" -> "we investigate *the* performance degradation from *a* model gradient perspective"; "we further define training loss" -> "we further define *the* training loss"; "where ... is prediction model ... CE is cross-entropy loss" -> "where ... is *a* prediction model ... CE is *the* cross-entropy loss".
	- Footnote 3 is very confusing; please consider rephrasing the entire footnote.
	- Lines 131-147 are also somewhat confusing and hard to comprehend due to language. Please consider rephrasing all of the paragraphs.
	- Line 148: "where gradient for model prediction is defines as" -> "where *the* gradient for *the* model prediction is *defined* as". Also, the inline *\frac* is not aesthetically pleasing, but that is a very minor complaint.
	- Line 161: Please consider either removing "*that*": "it is easy to obtain *that* the optimal dual variable v=..." -> "it is easy to obtain the optimal dual variable $v*$=..." or rephrasing to something more akin to "it is easy to see that the optimal dual variable is $v*$=..."
	- Please also consider rephrasing the next lines, from 162 to 164.
	- Missing commas on line 172: "the training loss for the whole training dataset, after model editing, is on par with (...)"
	- Please consider re-phrasing lines 199-200 to something more akin to "We randomly select a node from the validation set on which the well-trained model makes a wrong prediction."
- Both Tables 1 and 2 have results with different decimal precision for the standard deviation. For instance, GraphSAGE/GRE/CORA/DD has a single decimal ($3.36 \pm 0.2$) while GraphSAGE/GRE+/CORA/DD contains two decimals ($0.41\pm 0.07$). Please consider using the same amount of decimals for all of the results. It makes the table more aesthetically pleasing and easier to read.
- "OOM" is mentioned in the caption of Table 1 as being an out-of-memory error, but there are no OOMs in Table 1. However, Table 2 contains OOMs but the caption does not explain what OOM means.
- Please consider increasing the font of the text for both Figures 1 and 2.
- Figure 4, first row, first column: the figure covers half the "N" of "GCN". The plots are also not vertically aligned on the two columns.

Overall, I believe that the proposed method is solid due to the very good empirical results. However, the writing is lacking to a degree that makes the text very hard to follow and comprehend. I would encourage the authors to revise the text significantly, with multiple rounds of proofreading.

Due to concerns about the clarity and writing, I currently recommend rejection. However, I will revise my score if the authors significantly improve the writing during the rebuttal. I think that the overall work could potentially be impactful and useful from a practitioner's perspective.


[1]: https://arxiv.org/pdf/2305.15529

**Questions:**

Please see the weaknesses above.

**Limitations:**

The authors are discussing limitations in the appendix.

---

> ### Author Rebuttal · Authors · 2024-08-07
>
> Thanks for the constructive and insightful comments. We carefully revised the manuscripts based on all reviewers' comments. Please see the revised manuscript at https://anonymous.4open.science/r/Gradient_rewiring_editing-E16E/GRE_NeurIPS24.pdf.
>
> **[W1: Add comparison in terms of editing time and memory.]**
>
> Ans: For comparing time complexity and memory consumption, we measured the wall-clock editing time in milliseconds and GPU peak memory in megabytes, reporting average values over 50 independent edits in **Appendix D.5**.
>
> Our results show that the proposed method is scalable in terms of memory consumption and has manageable editing time overhead. For example, in the GraphSAGE architecture on the Flickr dataset, GRE+ (5) results in only a 13.5% increase in peak memory compared to GD. In terms of wall-clock editing time, the most time-consuming version, GRE+ (5), shows an insignificant overhead, with only a 6.31% increase on the ogbn-products dataset in the GraphSAGE architecture. These observations demonstrate the scalability of GRE+ for large datasets. It is noteworthy that model editing is usually efficient and fast, making slightly slower editing affordable while improving editing effectiveness.
>
> |                |       Editor        | Flickr ET (ms) | Flickr PM (MB) | Reddit ET (ms) | Reddit PM (MB) | ogbn-arxiv ET (ms) | ogbn-arxiv PM (MB) | ogbn-products ET (ms) | ogbn-products PM (MB) |
> |----------------|---------------------|----------------|----------------|----------------|----------------|--------------------|--------------------|-----------------------|-----------------------|
> | **GCN**        | **GD**              | 67.46          | 707.0          | 345.23         | 3244.8         | 94.58              | 786.2              | 2374.15               | 14701.7               |
> |                | **ENN**             | 109.82         | 666.8          | 405.24         | 3244.8         | 242.85             | 786.2              | --                    | OOM                   |
> |                | **GRE**             | 63.93          | 695.8          | 391.54         | 3491.3         | 84.74              | 956.9              | 2400.78               | 17336.6               |
> |                | **GRE+ (2)**        | 100.45         | 696.0          | 457.08         | 3493.2         | 121.11             | 957.8              | 2413.69               | 17338.7               |
> |                | **GRE+ (3)**        | 115.29         | 697.9          | 509.44         | 3493.9         | 131.06             | 957.9              | 2471.23               | 17338.9               |
> |                | **GRE+ (5)**        | 155.05         | 698.6          | 603.85         | 3495.6         | 162.24             | 958.3              | 2591.06               | 17339.2               |
> | **Graph-SAGE** | **GD**              | 117.74         | 843.0          | 1024.12        | 4416.53        | 107.63             | 891.3              | 2125.07               | 13832.2               |
> |                | **ENN**             | 134.50         | 843.0          | 2597.21        | 4416.5         | 277.29             | 891.3              | --                    | OOM                   |
> |                | **GRE**             | 116.03         | 952.4          | 1089.29        | 4955.4         | 100.09             | 1072.5             | 2132.02               | 16254.1               |
> |                | **GRE+ (2)**        | 167.17         | 954.5          | 1267.13        | 4959.0         | 136.28             | 1073.7             | 2135.88               | 16255.9               |
> |                | **GRE+ (3)**        | 176.66         | 955.5          | 1363.53        | 4960.7         | 154.29             | 1074.0             | 2211.63               | 16256.0               |
> |                | **GRE+ (5)**        | 219.81         | 957.5          | 1603.03        | 4964.2         | 180.73             | 1075.5             | 2275.72               | 16256.3               |
>
> **[W2: Tackle lots of writing/grammar issues]**
>
> Ans: Thanks for the careful review. We have revised the manuscripts significantly according to your comments.
>
> **[W3: Format issues on Table and Figures]**
>
> Ans: We have revised **the caption and uniform decimal precision in Tables 1 and 2, increased the font of the text for both Figures 1 and 2, and revised Figure 4**.

---

> ### Comment · Reviewer_KXu3 · 2024-08-09
>
> I thank the authors for their rebuttal. I believe that the authors have significantly improved the writing for their revision. The new time and memory comparison also strengthens the paper.
>
> Still, I would like to point out that the writing could still be improved:
>
> - Line 298: _in this experiment, we **scrutinize** the sensitivity of our proposed method_ - in my opinion, "scrutinize" is an odd term in this context. While not wrong, I believe that "analyze" would have been a better choice.
>
> - Line 21: I think the first sentence is still an odd way to present GNNs; "_integrating_" doesn't really fix the main issue. I would suggest that the authors revamp their first introduction paragraph entirely.
>
> - Line 28: "_In the ideal scenario, the promising property of tackling such errors would be threefold: ..._" - something along the lines of "_An ideal method that could tackle such errors would need to have the following properties: ..._" would be, in my opinion, much clearer.
>
> - There are still some decimal precision issues in the tables - for instance, in Tab. 1 the SR column contains both $1.0$ and $0.98$, Tab. 2 Column Flickr-ACC; 2-GCN-GD $11.0$ for the std but GCN-GRE $1.50$.
>
>
> Again, this is not a comprehensive list. I would recommend that the authors do some more passes through the manuscript for the next revision.
>
> Nevertheless, the writing has been significantly improved, and the method seems effective and practical - I am leaning towards acceptance right now, and have updated my score from a reject (3) to a borderline accept (5), with the presentation score going from poor (1) to fair (2).
>
> I will continue watching the rebuttal discussions, and might modify the score further depending on other comments.

---

> > ### Author Response · Authors · 2024-08-12
> > **Response to Reviewer KXu3**
> >
> > We sincerely thank the reviewer for the thorough review and for carefully considering not only our responses but also our interactions with other reviewers. In response to your suggestion, we have conducted a comprehensive proofreading of the manuscript, and the revised version is now available at https://anonymous.4open.science/r/Gradient_rewiring_editing-E16E/GRE_NeurIPS24.pdf.  In addition to improving the manuscript, we have also open-sourced our code at https://anonymous.4open.science/r/Gradient_rewiring_editing-E16E/README.md to enhance the reproducibility of our experiments. We kindly ask the reviewer to consider raising the score if all concerns have been satisfactorily addressed.

---

### Official Review · Reviewer_LHi3 · 2024-07-11

**Soundness:** 2
**Presentation:** 2
**Contribution:** 2
**Rating:** 5
**Confidence:** 3

**Summary:**

The work introduces a novel method called Gradient Rewiring (GRE) to address the challenge of editable training in graph neural networks. Traditional fine-tuning approaches often struggle with maintaining performance for both target and training nodes. GRE aims to overcome this limitation by rewiring gradients in a way that preserves locality, leading to improved performance for both types of nodes. The method is designed to enhance the training process in graph neural networks by effectively updating node representations while maintaining the network's overall structure.

**Strengths:**

1. The method of the paper is clear, intuitive, and easy to implement.

2. The writing is clear, making it easy to understand and read.

**Weaknesses:**

1. Although the experiments were conducted on graph datasets, the proposed method is not specifically designed for graphs. This general approach can be tested on various tasks.

2. There are noticeable formatting errors and typos in the paper.

**Questions:**

1. It would be beneficial to provide a more detailed explanation of the experimental process and results in Figure 1. In the first row of Figure 1, the images indicate that the gradients for testing and training become very similar, suggesting that the test loss and train loss should decrease simultaneously. However, in the second and third rows, the test loss decreases while the training loss increases. Please further explain this observation.

2. The training + rewriting pipeline is similar to curriculum learning, where the network is trained in two phases: first on simple samples, then on more difficult ones. I recommend the authors conceptually compare their method with curriculum learning. Additionally, I suggest experimenting with batch data rewriting, which might be more practical compared to rewriting individual samples or sample sequences.

**Limitations:**

The limitations has been discussed in the Appendix.

---

> ### Author Rebuttal · Authors · 2024-08-07
>
> Thanks for the constructive and insightful comments. We carefully revised the manuscripts based on all reviewers' comments. Please see the revised manuscript at https://anonymous.4open.science/r/Gradient_rewiring_editing-E16E/GRE_NeurIPS24.pdf.
>
> **[W1: This general approach can be tested on various tasks.]**
>
> Ans: Thank you for your insightful observation. We agree that while our experiments were conducted on graph datasets, the proposed gradient rewiring method is not inherently specific to graphs, the gradient rewiring method is particularly **suitable in the graph domain due to the small model size**. Specifically, graph models are typically a few layers and thus are smaller in model size compared to models (e.g., Transformers) used in NLP and CV tasks. This results in lower computational and storage costs for gradients, making our strategy particularly suitable for the graph domain. Additionally, it is more challenging to edit nodes in a graph due to the **inherent propagation process within neighborhoods**. Such propagation may lead to significant gradient discrepancies within the graph domain.
>
> In this work, we primarily focus on **gradient rewiring within the graph domain**, exploring multiple editing manners (i.e., independent editing, sequential editing, and batch editing). **We acknowledge the potential applicability of our gradient rewiring strategy to other domains and leave its exploration for future work in Appendix B**.
>
> **[W2: There are noticeable formatting errors and typos in the paper.]**
>
> Ans: We have carefully revised the manuscript. Please check the revised manuscript for the details.
>
> **[Q1: Provide a more detailed explanation of the experimental process and results in Figure 1.]**
>
> Ans: Thank you for the insightful comment. Gradient similarity does not necessarily imply consistent test and training loss. Loss consistency primarily depends on the model parameters rather than the first-order gradient. Therefore, there are cumulative effects of gradient inconsistency; significant initial gradient discrepancies can lead to substantial differences in model parameters, resulting in inconsistent training and test loss. We have added a more detailed explanation in Figure 1.
>
> **[Q2a: Conceptually compare their method with curriculum learning]**
>
> Ans: Thanks for the insightful comment. We have added the discussion with curriculum learning in **Appendix~F**. Here is the discussion:
>
> Curriculum learning and model editing are two distinct approaches in the field of machine learning. Curriculum learning is an approach where the network is trained in a structured manner, starting with simpler tasks and gradually introducing more complex ones. This method aims to improve the learning process by mimicking how humans learn. Model editing is a fast and efficient approach to patch the well-trained model prediction for several failed test cases. Although both are multi-stage training stages, there are several key differences:
> (1) **Goals**: Curriculum Learning aims to improve the overall learning process by structuring the training data in a way that mimics human learning. In contrast, model editing aims to make targeted adjustments to a pre-trained model to correct undesirable behaviors. (2) **Approach**: curriculum learning mainly focuses on the sequence and complexity of the training data. Model editing typically modifies the model's parameters or architecture to correct undesirable behavior goals.
> (3) **Additional information in the multi-stage process**. Model editing requires failure feedback for well-trained models as the target samples to patch, e.g., test failure cases after production is launched. In other words, such feedback can only be obtained after model pertaining. In curriculum learning, all information is given in multi-stage training. In summary, curriculum learning focuses on structuring the training process to improve overall learning, while model editing focuses on making targeted adjustments to a pre-trained model to correct specific behaviors. Both approaches can be complementary and used together to achieve better model performance.
>
> **[Q2b: More experimental results on batch editing.]**
>
> Ans: We have added experimental results on batch editing in **Appendix D.4 (Tables 6 and 7)**. We observe that, compared to independent editing, batch editing is more challenging as all batch samples need to be patched simultaneously. Additionally, GRE and GRE+ both demonstrate significant performance improvements compared to GD and ENN on various model architectures and datasets.

---

> > ### Author Response · Authors · 2024-08-12
> >
> > Thank you to the reviewer for the constructive comments and positive outlook on our paper. I wanted to kindly remind the reviewer that the discussion period is nearing its end, and I would greatly appreciate any additional feedback. Your insights have been invaluable in refining this work, and we are eager to address any remaining concerns before the final decision.

---

### Official Review · Reviewer_VRYA · 2024-07-15

**Soundness:** 3
**Presentation:** 2
**Contribution:** 3
**Rating:** 6
**Confidence:** 4

**Summary:**

This paper tackles the challenge of editing GNNs. The authors highlight a key issue in GNN editing: the gradient inconsistency between target and training nodes, which can degrade performance when the model is fine-tuned using only the target node’s loss. To address this, they introduce the Gradient Rewiring (GRE) method, which preserves the performance of training nodes by storing an anchor gradient and rewiring the target node’s loss gradient accordingly. The efficacy of GRE is validated through experiments across various GNN architectures and graph datasets, demonstrating its potential to enhance model adaptability without compromising existing accuracies.

**Strengths:**

- This paper is well-motivated and well-organized, addressing a research question that remains to be further investigated. The provided visualization also aids in understanding the motivations behind the study.
- The authors offer a detailed derivation process, making the methodology easy to follow.
- The method was tested on large-scale datasets, demonstrating the model's scalability.

**Weaknesses:**

- The significance of editing graph neural networks remains unclear. Editing seems to lead to significant performance degradation, especially on large-scale datasets. The authors should discuss why such significant efforts are warranted to modify predictions on individual samples.
- I recognize that considering graph neural network editing from the perspective of gradient rewriting is novel, but the authors have only discussed model editing in related work. The lack of gradient modification literature makes it hard to position this paper.
- Some experimental details are missing, such as dataset splitting. These details are crucial for evaluating the proposed method.
- The time complexity is not provided.
- Some minor issues (e.g., a superscript linking to nothing in the caption of Figure 1).

**Questions:**

- In what scenarios would people be willing to accept a substantial overall performance drop in order to modify predictions for individual samples?
- Are there any methods developed for editing neural networks on non-graph data based on gradient modification? Or are there any works on gradient modification that are relevant to the method in this paper?

**Limitations:**

As the authors discussed in Appendix B.

---

> ### Author Rebuttal · Authors · 2024-08-07
>
> Thanks for the constructive and insightful comments. We carefully revised the manuscripts based on all reviewers' comments. Please see the revised manuscript at https://anonymous.4open.science/r/Gradient_rewiring_editing-E16E/GRE_NeurIPS24.pdf.
>
> **[W1: The significance of editing graph neural networks remains unclear]**
>
> Ans: Model editing is crucial for ensuring that machine learning models remain reliable and effective post-deployment. This practice addresses high-profile failure cases and misbehaviors that emerge after a model's initial training, often brought to attention through user feedback.
>
> In general, there are **editing-worthy scenarios, such as high-profile failures and prioritizing critical mistakes**. For example, users may try to prompt large language models (LLMs) to misbehave (e.g., giving criminal advice, system prompt, etc.) to induce "high-profile failures." In the context of self-driving cars, misclassifying a child as a cat poses far greater risks than misclassifying a cat as a dog. Although graph applications typically don't have this kind of direct interaction with users and are not as intuitive, they certainly have high-stake scenarios, such as **patient readmission [1] and flood prediction [2]**, which warrant the study of model editing.
>
> In short, while model editing has been widely explored in computer vision (CV) and natural language processing (NLP) tasks, it has rarely captured attention in the graph learning community. It is indispensable to investigate graph model editing for high-stake graph applications.
>
> [1] Predicting patient readmission risk from medical text via knowledge graph enhanced multiview graph convolution. SIGIR 2021
> [2] Kazadi, Arnold N., et al. "Flood prediction with graph neural networks." Climate Change AI. Climate Change AI (2022).
>
> **[W2: Please discuss gradient modification literature in related work]**
>
> Ans: We have added a discussion on the literature regarding gradient modification in continual learning and meta-learning. Please see more details in Appendix~E.
>
>
> **[W3: More experimental details such as data spliting]**
>
> Ans: For all datasets, we first randomly split the data into train, validation, and test sets. Specifically, we ensure that each class has 20 samples in the training set and 30 samples in the validation set. The remaining samples are used for the test set. The target node is randomly selected multiple times from the validation set where the well-trained model makes incorrect predictions.
>
>
> **[W4: The time complexity is not provided.]**
>
> Ans: For comparing time complexity and memory consumption, we measured the wall-clock editing time in milliseconds and GPU peak memory in megabytes, reporting average values over 50 independent edits in **Appendix D.5**. Due to the limited rebuttal length, we only show time complexity and memory consumption for GCN.
>
> Our results show that the proposed method is scalable in terms of memory consumption and has manageable editing time overhead. For example, in the GraphSAGE architecture on the Flickr dataset, GRE+ (5) results in only a 13.5% increase in peak memory compared to GD. In terms of wall-clock editing time, the most time-consuming version, GRE+ (5), shows an insignificant overhead, with only a 6.31% increase on the ogbn-products dataset in the GraphSAGE architecture. These observations demonstrate the scalability of GRE+ for large datasets. It is noteworthy that model editing is usually efficient and fast, making slightly slower editing affordable while improving editing effectiveness.
>
> |                |       Editor        | Flickr ET (ms) | Flickr PM (MB) | Reddit ET (ms) | Reddit PM (MB) | ogbn-arxiv ET (ms) | ogbn-arxiv PM (MB) | ogbn-products ET (ms) | ogbn-products PM (MB) |
> |----------------|---------------------|----------------|----------------|----------------|----------------|--------------------|--------------------|-----------------------|-----------------------|
> | **GCN**        | **GD**              | 67.46          | 707.0          | 345.23         | 3244.8         | 94.58              | 786.2              | 2374.15               | 14701.7               |
> |                | **ENN**             | 109.82         | 666.8          | 405.24         | 3244.8         | 242.85             | 786.2              | --                    | OOM                   |
> |                | **GRE**             | 63.93          | 695.8          | 391.54         | 3491.3         | 84.74              | 956.9              | 2400.78               | 17336.6               |
> |                | **GRE+ (2)**        | 100.45         | 696.0          | 457.08         | 3493.2         | 121.11             | 957.8              | 2413.69               | 17338.7               |
> |                | **GRE+ (3)**        | 115.29         | 697.9          | 509.44         | 3493.9         | 131.06             | 957.9              | 2471.23               | 17338.9               |
> |                | **GRE+ (5)**        | 155.05         | 698.6          | 603.85         | 3495.6         | 162.24             | 958.3              | 2591.06               | 17339.2               |
>
>
>
> **[W5: Some minor issues (e.g., a superscript linking to nothing in the caption of Figure 1).]**
>
> Ans: We have carefully revised the manuscript. Please check the revised manuscript for the details.

---

> > ### Comment · Reviewer_VRYA · 2024-08-11
> >
> > Thanks for the clarifications. Most of my concerns have been addressed. However, I would suggest releasing the code in the discussion phase to make your experiments convincing. For now, I maintain my original score.

---

> > > ### Author Response · Authors · 2024-08-12
> > > **Response to Reviewer VRYA**
> > >
> > > Thank you once again for your constructive feedback. We are pleased that our response has addressed most of your concerns. In line with your suggestion, we have now open-sourced our code at https://anonymous.4open.science/r/Gradient_rewiring_editing-E16E/README.md. We hope that this will enhance the transparency and reproducibility of our experiments, and we kindly ask the reviewer to consider raising the score if all concerns have been satisfactorily addressed.

---

> > > > ### Comment · Reviewer_VRYA · 2024-08-12
> > > >
> > > > I appreciate the implementation details provided by the authors. After a review of the code, I have raised the score to weak accept.

---

> > > > > ### Author Response · Authors · 2024-08-12
> > > > >
> > > > > Thank you once again for your constructive comments and for taking the time to review our implementation details and code. We appreciate the productive discussion and are pleased that you have decided to raise the score.

---

### Author Rebuttal · Authors · 2024-08-07

# A summary of our rebuttal.
We thank all reviewers for your valuable time and comments. We are glad that many reviewers found the following:
* **Our paper is well-motivated and well-organized**
     * R `VRYA`: This paper is well-motivated and well-organized, addressing a research question that remains to be further investigated.
     * R `KXu3`: I believe the problem to be an important practical problem and that the paper is well-motivated. The main observation of the paper, i.e., that the gradients of the target and the training nodes are inconsistent, is interesting and novel in the context of Graph Neural Networks.
     * R `X56x`: The introduction and motivation of the paper are generally well-written and easy to follow.
- **Our method is clear, intuitive, and easy to follow.**
     * R `VRYA`: The authors offer a detailed derivation process, making the methodology easy to follow.
     * R `LHi3`: The method of the paper is clear, intuitive, and easy to implement.
- **The experimental results are solid and strong.**
     * R `VRYA`: The method was tested on large-scale datasets, demonstrating the model's scalability.
     * R`KXu3`: The proposed method seems to perform well consistently...., and I believe the results are solid.
     * R`X56x`: In most situations, the proposed GRE+ proves to be the best method.

On the other hand, aside from some cosmetic suggestions, the reviewers have brought up the following suggestions:
- **Editing time and peak memory comparison** (R `VRYA`, `KXu3`, `X56x`): We conduct editing time and peak memory comparison in ** Appendix D.5**.
- The **significance of editing graph neural networks** remains unclear (R `VRYA`): there are multiple editing-worthy scenarios, such as high-profile failures and prioritizing critical mistakes. Two specific applications such as patient readmission and flood prediction.
- **Batch editing experimental results** (R `X56x`): The batch editing results are shown in **Appendix D.4** (Tables 6 and 7).
- ENN results in sequential editing (R `X56x`): The test accuracy comparison results are shown in **Appendix D.6** (Figure 6).
- This general approach can be tested on various tasks (R `LHi3`):  (1) the gradient rewiring method is particularly suitable in the graph domain due to the small model size. (2) it is more challenging to edit nodes in a graph due to the inherent propagation process within neighborhoods. (3) acknowledge the potential applicability of our gradient rewiring strategy to other domains in **Appendix B**.
- Related work on gradient modification (R `VRYA`): We add Related work on gradient modification in **Appendix E**.
- Detailed explanation for Figure 1 (R `LHi3`): We add a more detailed discussion in Section 3.1.
- Discussion on the differences with curriculum learning (R `LHi3`): We have added the discussion with curriculum learning in **Appendix~F**.

Based on all reviewers' comments, we carefully revised the manuscript. Please see the revised manuscript at https://anonymous.4open.science/r/Gradient_rewiring_editing-E16E/GRE_NeurIPS24.pdf.

---

### Comment · Area_Chair_tZXu · 2024-08-08

Dear Reviewers,

Please read authors' responses carefully and provide your answers.

Thanks,
AC

---

### Decision · Program_Chairs · 2024-09-25

**Decision:**

Accept (poster)

**Comment:**

This paper proposes a method to simply rewire (modify) the gradient for the target loss to maintain the overall performance during the model editing of GNNs. In specific, it proposes a novel constrained optimization problem to regulate the change of the training loss after the model editing, and solves it efficiently by the gradient rewiring using the anchor gradient of the overall training loss. Experimental results on various graph datasets show that the proposed method, called GRE(+), improves performances over baselines in reducing the overall accuracy dropdown(drawdown?) while maintaining the high target success rate.

Overall, the motivation is clear, and the proposed method seems to be technically sound, supported by extensive empirical validation. The authors also thoroughly address many concerns and issues raised by the reviewers, and the reviewers’ scores are one weak accept and three borderline accepts.

However, the description of the proposed method and its writing still need to be improved. Especially, the derivation of the proposed constrained optimization is somewhat unclear. Why do we have such two constraints as Eq. (3),(4)? What if we just take one of them? What if we take an average of prediction differences in Eq. (4)? Even, there is no ablation study on this. In addition, what is \theta in Eq. (2) different from \theta’?, and how can we derive the approximation in Eq. (6)? Also, figures and tables should be placed close to their descriptions. Especially, Table 1 and 2 in the middle of Method Section disrupt clearly understanding the explanation of the method.

In conclusion, this paper is a borderline paper and has been improved significantly through the rebuttal. Based on the consensus among the reviewers, I would recommend the paper to be accepted. However, the authors should revise the description of the proposed method more clearly, and more explanation and ablation studies regarding the proposed optimization problem should be conducted.